# Plant interactions associated with a directional shift in the richness range size relationship during the Glacial-Holocene transition in the Arctic

Ying Liu [1,2], Simeon Lisovski [1], Jérémy Courtin [1], Kathleen R. Stoof-Leichsenring [1] & Ulrike Herzschuh [1,2,3] ✉

A nearly ubiquitous negative relationship between taxonomic richness and mean range-size (average area of taxa) is observed across space. However, the complexity of the mechanism limits its applicability for conservation or range prediction. We explore whether the relationship holds over time, and whether plant speciation, environmental heterogeneity, or plant interactions are major factors of the relationship within northeast Siberia and Alaska. By analysing sedimentary ancient DNA from seven lakes, we reconstruct plant richness, biotic environmental heterogeneity, and mean range-size over the last 30,000 years. We find positive richness to range-size relationships during the glacial period, shifting to negative during the interglacial period. Our results indicate neither speciation nor environmental heterogeneity is the principal driver. Network analyses show more positive interactions during the glacial period, which may contribute to positive richness to range-size relationships. Conversely, in the interglacial environment, negative interactions may result in negative relationships. Our findings suggest potential susceptibility to invasion but conservation advantages in far northern tundra given their positive interactions.

As a universal characteristic of every species, geographic range-size is a strong predictor of species expansion and vulnerability to extinction, and can be regarded as an indicator for species conservation status. A negative relationship between richness and mean range-size is nearly ubiquitously observed across space for various taxonomic groups[1–3], yet the exact mechanism of the richness to mean range-size relationship is still poorly understood. Furthermore, it remains unclear whether the spatial relationship holds true in the temporal domain. It is of particular interest whether the observed species richness decline in the context of global change[4] is associated with an increase in mean range-size. Alternatively, do changes in species range-size result in

richness adjustments? Answering these questions requires the knowledge to exploit the richness to range-size relationship for predictions, for example, in the context of conservation efforts[5,6].

Based on previous research, three main hypotheses are proposed to explain the widely observed negative richness to range-size relationship. First, long-term (millions of years) temperature-related speciation and species extinction may result in negative richness to mean range-size relationships in the spatial domain[7,8]. Higher temperatures increase metabolic rates and promote speciation rates[9]. Within a region with high speciation rates, new species are initially constrained within a small range, resulting in regions with high species richness and

[1]Alfred Wegener Institute Helmholtz Centre for Polar and Marine Research, Polar Terrestrial Environmental Systems, 14473 Potsdam, Germany. [2]Institute of Environmental Science and Geography, University of Potsdam, 14476 Potsdam, Germany. [3]Institute of Biochemistry and Biology, University of Potsdam, 14476 Potsdam, Germany. ✉e-mail: Ulrike.Herzschuh@awi.de

smaller range-sizes[8,10] as exemplified by the equatorial region[11]. In contrast, low temperatures and recurrent extensive glaciation in high-latitude regions have resulted in less speciation and increased the risk of regional species extinctions[12]. Thus, variation in speciation and extinction rates along the temperature gradient results in a negative richness to range-size relationship in space (Fig. 1a)[7,8]. However, this hypothesis applies to the global pattern; it is not solely attributed to temperature differences across space but also to the temporal differences in species accumulation and extinction over millions of years. Within a specific region and over a millennial time scale, the impact of speciation and extinction may be limited. Hence, to shift the perspective to a temporal domain over millennia, richness and range changes are not driven by speciation. Moreover, without considering extinction, there is no expected relationship between richness and range-size across time (Fig. 1b).

Second, environmental heterogeneity can result in a negative richness to mean range-size relationship[13,14]. Heterogeneous environments, for example complex forest ecosystems, may promote the colonisation of specialised taxa that can coexist[15,16]. This leads to a high

richness of taxa with narrow ecological niches and limited geographical ranges[13], resulting in a negative richness to range-size relationship along spatial heterogeneity gradients (Fig. 1c). Accordingly, in the temporal domain, a stable negative richness to mean range size relationship will be expected (Fig. 1d). For example, the diversity decline related to the expansion of generalist taxa with a large distribution range can be explained by human landscape homogenisation[17,18].

Third, biotic interactions among taxa may drive richness to range-size relationships[3]. However, there is a lack of systematic investigations along spatial gradients. The stress-gradient hypothesis[19,20] suggests that in less stressful environments, negative interactions are promoted, while in stressful environments, positive interactions should be more common. In line with the stress-gradient hypothesis, it may be hypothesised that in less stressful environments, high richness can lead to increased competition and predation (negative interactions), limiting growth rate[21] and reducing range-sizes[3], resulting in a negative richness to-range-size relationship. In contrast, environmental stress can promote positive interactions among taxa allowing for mutually

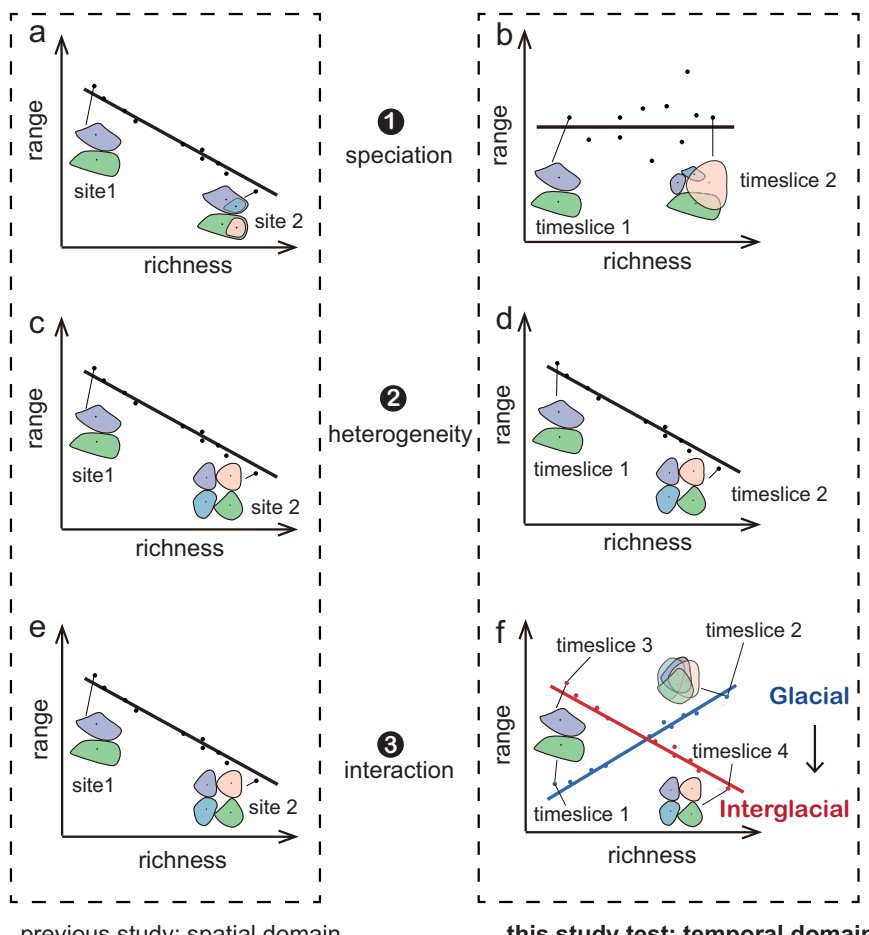

previous study: spatial domain                                              this study test: temporal domain

**Fig. 1 | Evaluating three hypotheses of how species richness relates to range-size over space (previous studies[3,7,8,13,14]) and time (this study), using a hypothetical richness increase scenario.** Coloured polygons represent the distribution range of different plant taxa. Under the speciation hypothesis, regions experiencing high speciation rates over millions of years generate taxa with a constrained range, resulting in a negative richness to range-size relationship (**a**), in the temporal domain, no speciation occurs on millennial time scales, neither during the glacial period nor the interglacial period. The richness increases are not driven by speciation, and range-size remains stable or changes without a discernible pattern, thus, there is no richness to range-size relationship (**b**). Under the environmental heterogeneity hypothesis, the temporal domain pattern (**d**) mirrors the spatial

domain pattern (**c**). Spanning both glacial and interglacial periods, heterogeneous environments favour a high richness of taxa with narrow ecological niches and limited geographic ranges, leading to a stable negative richness to range-size relationship (**d**). Under the plant interaction hypothesis, widespread negative richness to range-sizes relationships are observed in the spatial domain, attributed to negative interactions (**e**). Unlike the spatial domain result, in the temporal domain, a positive interaction during the glacial period expands taxa range-sizes (**f**), resulting in a positive richness to range-size relationship; while a negative interaction during the interglacial period constrains the range-sizes of the taxa, resulting in a negative relationship. The relationship shifts from positive during the glacial towards negative during the interglacial (**f**).

wide ranges[19,20,22,23], which would be reflected by a positive richness to range-size relationship. However, positive richness to range-size relationships are rarely detected in space, which may be because other factors obscure the relationship when multiple drivers are correlated. In the temporal domain, an environmental stress release may lead to a shift from a positive to a negative richness to range-size relationship (Fig. 1). For example, this shift might occur when transitioning from the stressful tundra environment during the glacial period to the less stressful taiga environment in the Holocene[24–26] (Fig. 1f).

To disentangle the potential factors causing richness to range-size relationships, conducting studies from a temporal perspective within confined geographical regions and appropriate palaeoenvironmental settings is essential. For this purpose, northeast Siberia plus Alaska is a suitable study area. Because of polar amplification, this high-latitude area has experienced strong changes in temperature on millennial time scales[23], mirrored by ecological turnover along with treeline translocation[27]. However, unlike other climate-sensitive high-latitude regions, this area was not covered by ice shields during the last glacial period[28,29]. As such, lagged species assembly effects in the course of new habitat formation after deglaciation[30] are minimal due to the potential for regional recruitment. The area also provides high-quality lake sedimentary archives reaching back to the Last Glacial Maximum (LGM; 21 ka BP) and beyond. Furthermore, previous studies from the area have yielded high-quality sedimentary ancient DNA (sedaDNA) records. Compared to traditional pollen analysis, sedaDNA can identify past plants to a higher taxonomic resolution, which allows the reconstruction of past plant richness and distribution changes at a reasonable taxonomic level[31].

Here, we use sedaDNA-based plant community reconstructions of the last 30,000 years to investigate richness to mean range-size relationships in the temporal domain within northeast Siberia plus Alaska (Fig. 1). Our straightforward study design will test the following hypotheses. (1) If millions of years of speciation and species loss are the primary factors, when applying this pattern to millennial studies, especially after eliminating the potential impact of extinction, an insignificant richness to range-size relationship would be expected through time. In contrast, if (2) heterogeneity dominates, which can change on millennial time scales, we expect that higher species richness is consistently related to smaller range-size over time. Alternatively, (3) along with postglacial forest expansion, interactions might have changed from a system that is dominated by mutualistic (positive) interactions to systems dominated by competitive (negative) interactions, which should be reflected by a shift from a positive to a negative richness to mean range-size relationship. Overall, our study aims to disentangle the potential factors, providing insight into temporal plant richness and mean range-size relationships to inform future conservation and protection.

## Results and Discussion
### Plant richness to range-size relationships from sedaDNA records
The study analysed samples from seven lake sediment records from northeast Siberia and Alaska (Fig. 2a), most covering the past 30,000 years at a millennial resolution (Supplementary Data 1). Sedimentary ancient DNA was analysed using the metabarcoding method targeting the P6 loop of the *trn*L (UAA) intron, and taxonomically assigned with the customised regional SibAla_2023 database[32]. The database has a taxonomic coverage of 95.7% at the family level, 89.4% at the genus level, and 70.1% at the species level, compared to occurrences in the Global Biodiversity Information Facility (GBIF)[33]. The final dataset used in this study comprises 352 samples, generating 70,675,012 reads of 625 Amplicon Sequence Variants (ASVs) with 100% assignment. Total richness was calculated by counting taxa from all records for 1000-year time-slices for the last 30,000 years and averaging over 5000-year moving time-windows. Range-size was calculated from lake-wise resampled data by counting the lake number of each taxon's

occurrence (AOO: Area of Occupancy method), and the area polygon covering all the lakes of taxa occurrence (EOO: Extent of Occurrence), respectively. The mean range-size ("range-size" for simplicity) refers to the average range-size of all taxa in every 1000-year time-slice. The findings from both approaches exhibited concordance (Supplementary Note 3): the AOO result is shown in the text (Fig. 3), while the EOO is shown in the Supplementary files (Supplementary Figs. 1, 2, 3).

In accordance with expectation, we find a significant negative spatial relationship between sedaDNA plant taxa richness and range-size for the seven lakes for the modern time-slice (r = -0.72, $p = 2.2 \times 10^{-16}$) (Fig. 2b). Modern plant information for the northeast Siberia and Alaska region was derived from GBIF, and range-size was determined based on the sum of occupied 200 km x 200 km grid cells. In space, plant richness and range-size among grid cells show negative relationships (r = -0.37, $p = 1.1 \times 10^{-13}$) (Fig. 2c), as observed in the DNA results (Fig. 2b). Furthermore, for the same GBIF-derived taxa, their range-sizes were calculated using the AOO method, based on grid cells centred around the seven lakes, and the larger Siberian and Alaskan regions. The results based on two regions reveal a high correspondence pattern (Fig. 2d), validating the representation of plant taxa range-sizes based on the seven lakes. Thus, our proxy-data results agree with modern observations from the region, which, in turn, is consistent with the negative plant richness to range-size relationship observed in many other studies worldwide at different spatial scales[3,34].

The 30-ka time-series of all-record total richness starts at a medium level, has high values before and around the LGM (21 ka BP), a minimum during the early late glacial (19–15 ka), and overall high values during the Holocene with maxima in the early (11–7 ka) and late Holocene (5–1 ka) (Fig. 3a). The range-size time-series starts with the highest value (30–26 ka), decreases continuously until the LGM (21 ka BP), reaches a minimum during the early late glacial (19–15 ka), a relatively low value during the early Holocene (14–8 ka), and increases until the late Holocene (6–2 ka) with a second maximum (Fig. 3b).

Correlation analyses find positive temporal relationships between plant richness and range-size within 5000-year time-windows for the glacial period (30–11 ka) and negative relationships for the Holocene (11–0 ka, Fig. 3c, Fig. 4). All time-window correlations are significant ($p < 0.001$), except for the time-slice 13–9 ka (see Supplementary Table 1 for details), which possibly reflects the overlapping signals of the glacial and Holocene periods. Our results fundamentally differ from the widespread negative richness to range-size relationships previously reported for the spatial domain[3,34].

### Potential temporal factors of the past richness to range-size relationship
Given that we detect significant richness to range-size relationships for almost all time-windows in our study over the last 30,000 years, and that the potential extinction effect was eliminated by using taxa with 100% assignment to the "SibAla_2023" database, we conclude that processes operating on millennial time-scales also influence the temporal relationship between richness and mean range size. We reject hypothesis (1), which posited that temperature-related speciation and glaciation-related species extinctions are the sole major factor shaping the richness to range-size relationship. The pollen-based regional temperature increases markedly from the glacial to the Holocene in accordance with Northern Hemisphere trends (Fig. 3g, h)[23]. Higher temperature has also been acknowledged to stimulate metabolism and increase the speciation rate and biodiversity, exemplified by the richness decrease from the equator to poles[35]. Accordingly, speciation processes may have increased during the Holocene in the study region. However, the period of our study is too short to accumulate sufficient mutations for speciation which requires millions of years[36], and the effect of extinction was eliminated by using taxa with 100% assignment to the "SibAla_2023" database (Supplementary Note 5). Therefore,

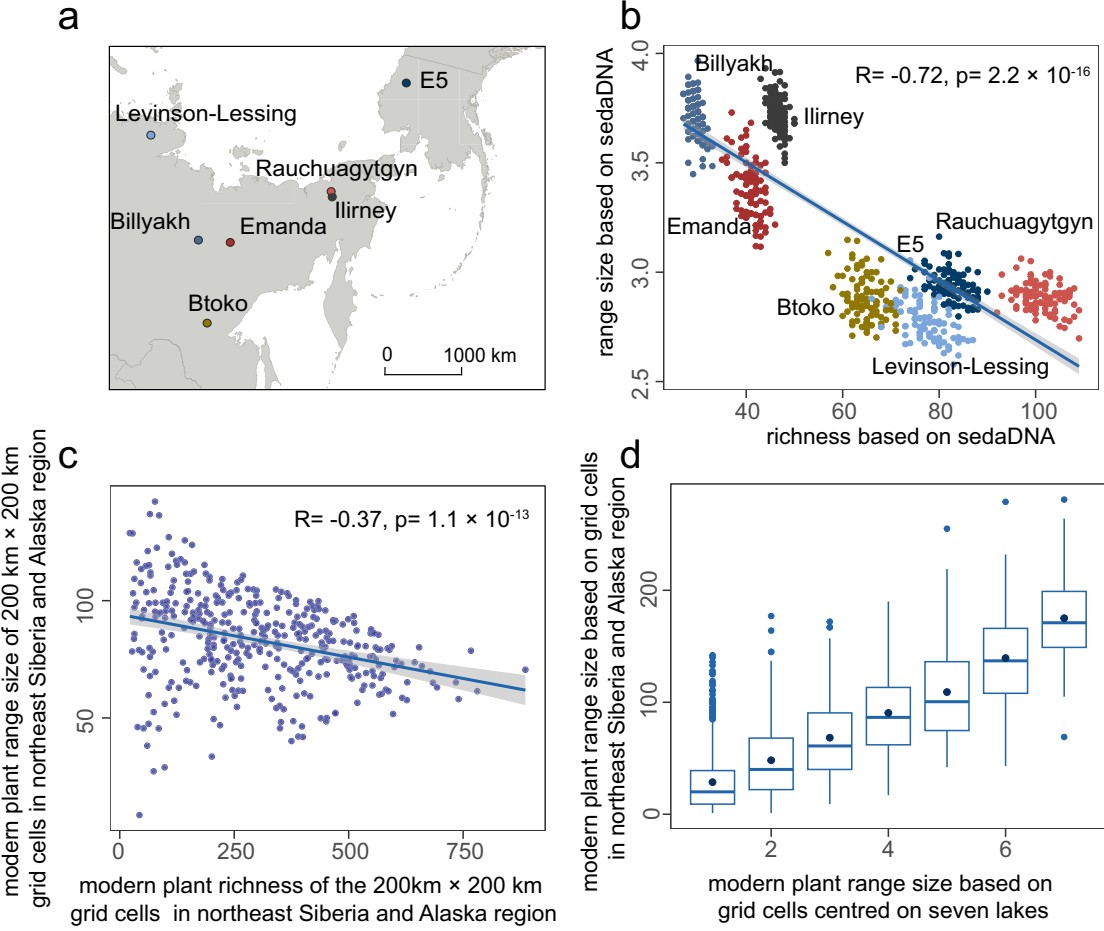

**Fig. 2 | Plant richness to range-size relationship across space. a** Location of the seven lakes in northeast Siberia and Alaska (map created using the Free and Open Source QGIS); **b** spatial domain plant richness to range-size relationship based on the modern time-slice (2000–0 years) from sedimentary ancient DNA. Different coloured points represent the richness and range-size values for each of the lakes. A linear regression is fitted to the points with a confidence interval of 95% and a Spearman's rank correlation coefficient was calculated to assess the relationship between plant richness and range-size, with a p-value of $2.2 \times 10^{-16}$; **c** plant richness to range-size relationship based on modern plant taxa occurrence in 200 km × 200 km grid cells in northeast Siberia and Alaska. Each point represents the plant richness and range-size for each grid cell. A linear regression is fitted to the points with a 95% confidence interval, and a Spearman's rank correlation

coefficient was calculated to assess the relationship between plant richness and range size, with a p-value of $1.1 \times 10^{-13}$; **d** comparison of the range-sizes of the same taxa (present in both the 200 km x 200 km grid cells centred on the seven lakes and the 200 km x 200 km grid cells within the northeast Siberia and Alaska region) based on two methods: summing the grid cells centred on the seven lakes, and summing the grid cells within the northeast Siberia and Alaska region. The total taxa number is 1249, the middle line in the box corresponds to the median value, the middle point in the box corresponds to the mean value, the edges of the box correspond to $25^{th}$ (lower edge) and $75^{th}$ (upper edge) percentile, and the ends of the whiskers correspond to the minimum and maximum range-size based on the 200 km x 200 km grid cells centred on the seven lakes. Source data are provided as a Source Data file.

speciation and extinction are not the primary drivers of the temporal richness to range-size relationship on millennial time scales.

Since we find a flip in the temporal richness to mean range-size relationship (Figs. 3c, 4), heterogeneity cannot represent the main driver of the relationship throughout the investigated period; refuting hypothesis (2). We find low heterogeneity during the late marine isotope stage (MIS) 3 (30–24 ka) when using plant community beta-diversity between records as a proxy (Fig. 3d). This confirms previous findings that taxa of the Eurasian mammoth steppe had a widespread distribution[37]. Although at the site scale this period may have the highest richness compared to other periods[31], at the regional scale, the low heterogeneity supports relatively low overall richness. The highest values of heterogeneity were during the early Holocene when there was widespread forest expansion in response to warming[27]. This may have resulted in site-specific forest trajectories[38] and temporal vegetation–climate disequilibrium[39], which may have supported the high early-Holocene richness maximum but does not align with the minimum range-size. Thus, our results do not support the proposition

that high heterogeneity raises species richness and reduces the availability of habitats by increasing the niche dimensionality, as stated by Stein and Kreft[40]. Rather, the area-heterogeneity trade-off hypothesis indicates that a decline in available habitats for taxa will cause the mean population size to decrease and increase the probability of stochastic extinctions, thereby reducing species richness[13,41].

Serving as nurse plants, cushion plants inhabiting arctic or alpine regions exhibit a low-growing, mat-forming growth pattern, which supports a positive interaction with other plants[42], and a high proportion of cushion plants may lead to a more positive richness to range-size relationship through time. To test the plant interaction hypothesis, two generalised linear models were constructed, with richness to range-size relationship as binomial response variables (positive for the correlation $R > 0.2$, and negative for the correlation $R < -0.2$) and cushion plant abundance and tree abundance as the explanatory variables (Fig. 5). With increasing tree abundance in the area, the richness to range-size slope shifts from positive to negative (slope = -0.381, $p = 2 \times 10^{-16}$), and cushion plant abundance decreases

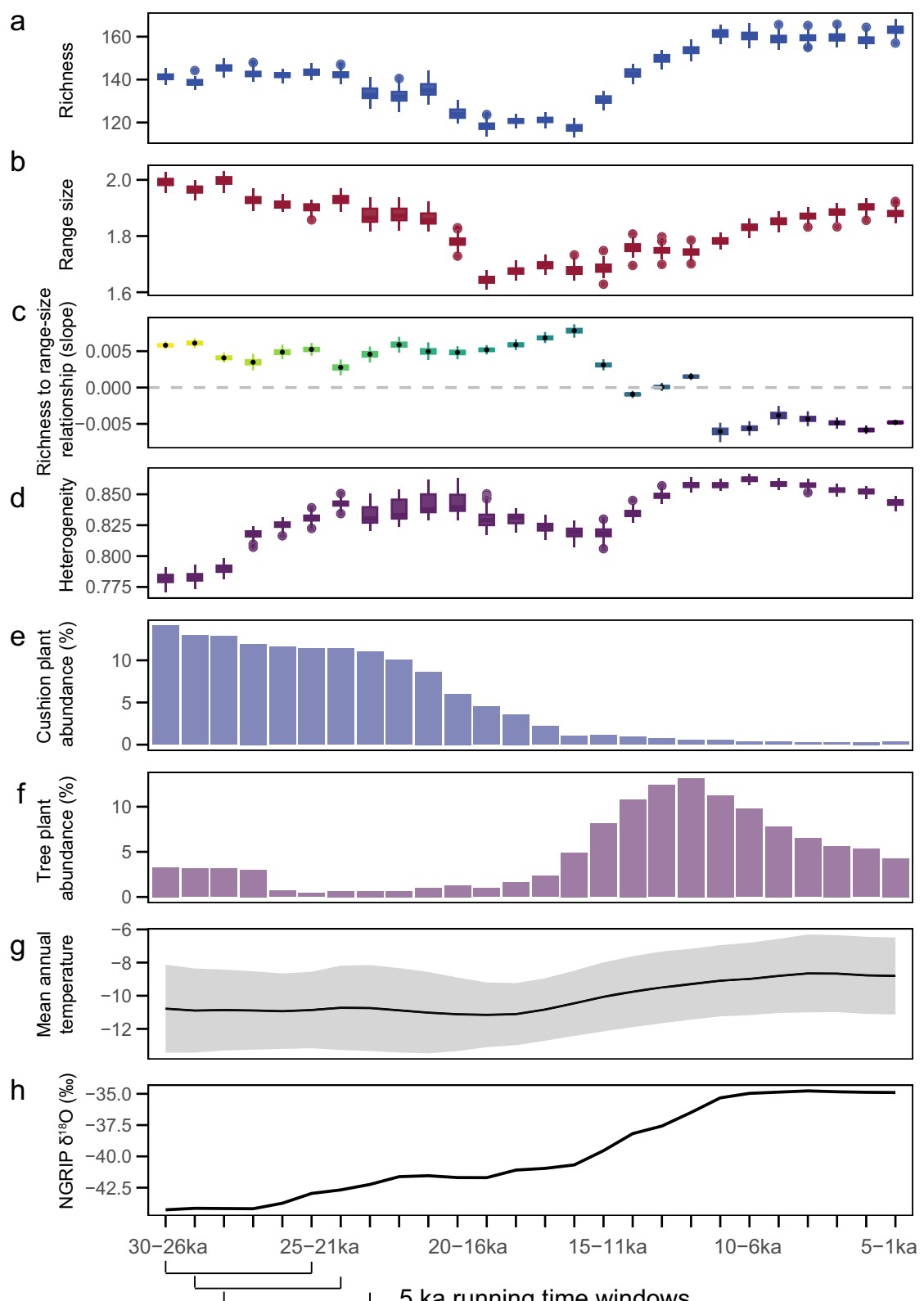

(slope=1.144, $p = 2 \times 10^{-16}$) (Figs. 3e, 5), which could be attributed to a positive interaction of cushion plants[42].

The network, constructed from positive pairwise plant taxa correlations (r > 0.6), delineates two communities (Fig. 6). The glacial community, incorporating cushion plant taxa, comprises 11 nodes and 30 positive internal edges; a 68.18% positive link proportion. In contrast, the Holocene community, which includes tree taxa, also consists

of 11 nodes, but 17 positive internal edges; a 38.64% positive link proportion. For the glacial community, more links are detected compared with the Holocene community, which indicates more associations among plant taxa and thus a higher possibility of positive plant interactions.

The cushion plants include some species in our dataset belonging to Saxifragaceae (*Saxifraga*), Caryophyllaceae (*Silene*, *Stellaria*),

**Fig. 3 | Changes in the plant richness to range-size relationship and potential drivers of the relationship. a** Plant richness change over time; **b** range-size change over time; **c**, the plant richness to range-size relationship shifts from positive to negative at the transition from the glacial to interglacial (the relationship was calculated based on the richness and range-size, colour gradient is similar to that in Fig. 4); **d** biotic environmental heterogeneity over time; **e** cushion plant abundance (%) over time based on the mean value of 100 resampling results of our sedimentary ancient DNA dataset from seven lakes in northeast Siberia and Alaska; **f** tree plant abundance (%) over time based on the mean value of 100 resampling results of our sedimentary ancient DNA dataset from seven lakes in northeast Siberia and Alaska; **g** reconstructed mean annual temperature (°C; with 95% confidence intervals) based on ten pollen sites in the study region using the weighted averaging partial least squares (WAPLS) method, over the past 30,000 years. Some sites are the same as the sedaDNA data sites. **h** temperature index represented by oxygen isotope values (δ¹⁸O) over the past 30,000 years from the North Greenland Ice Core Project

(GRIP)[24], higher values correspond to higher temperatures. To compare with the richness to range-size relationship (calculated based on 5,000-year (5 ka) intervals for each time-window), and to smooth the data to eliminate noise, all the data are shown in 5000-year running time-windows. For panels **a**, **b**, and **d**, the boxplot is based on the result of 100 resampling iterations, where the middle line in the box corresponds to the median value, the edges of the box correspond to 25th (lower edge) and 75th (upper edge) percentile, and the ends of the whiskers correspond to the minimum and maximum values for each time-window. For panel **c** the boxplot shows the modelling slope values for each time-window. Sample sizes are $n = 500$ for most windows, except "23–19ka", "22–18ka", "21–17ka", "20–16ka", and "19–15ka", which have $n = 474$. The boxplot's median represents the mean value, the box edges indicate the 25th and 75th percentiles, and the whiskers show the minimum and maximum values for each time-window. Source data are available in the Source Data file.

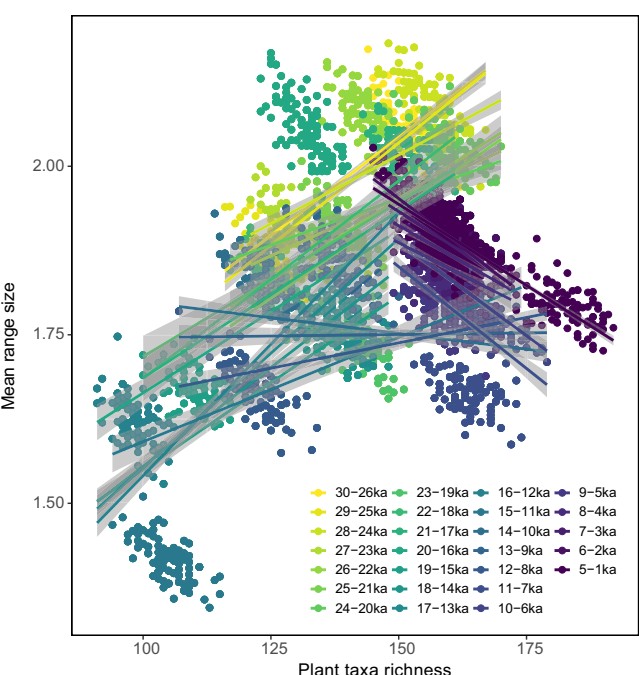

**Fig. 4 | Plant taxa richness to range-size relationships (with 95% confidence intervals) per 5000-year time-window over the last 30,000 years inferred from lake sediment ancient DNA collected from northeast Siberia and Alaska.** Coloured points show the richness and mean range-sizes of the 1000-year time-slice samples (100 resampling iterations). The mean range-size is determined by calculating the average number of lakes occupied. Source data are provided as a Source Data file.

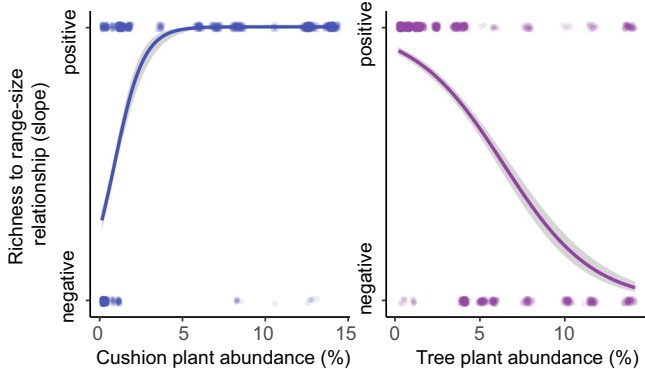

**Fig. 5 | Relationship between the richness to mean range-size relationship and taxa abundance.** Binomial regression was performed to examine the richness to mean range-size relationship (R > 0.2 for positive relationship, R < -0.2 for negative relationship) with relative abundance of cushion plant taxa (%) ($p$ value: $2 \times 10^{-16}$) (left panel) and tree plant taxa (%) ($p$value: $2 \times 10^{-16}$) (right panel), based on the median value of a 5,000-year running time-window for every resampling round. Source data are provided as a Source Data file.

Boraginaceae (*Eritrichium*), and Brassicaceae (*Draba*). In line with the network result, their ability to modify the physical environment by reducing wind speed, maintaining temperatures, retaining soil water, and keeping higher soil fertility[43–45], creates suitable habitat patches for other plant species and widens the ranges of coexisting species[46,47]. Additionally, *Saxifraga* has been demonstrated to facilitate pollination visitation and dispersal, indirectly contributing to range-size expansion[48]. Accordingly, higher cushion plant abundance during the glacial period represents an environment dominated by positive interactions, supporting a positive richness to mean range-size relationship.

Conversely, the absence of positive interactions with cushion plants could constrain a species' range. Woody taxa such as *Betula*, *Alnus*, *Salix*, *Populus*, and *Larix* in our study region, are tall and have dense canopy structures. This leads to a decrease in the amount of solar radiation reaching understorey plant taxa, which potentially

increases competition for light among species and decreases the understorey species richness[49]. Additionally, their deciduousness contributes to rapid resource acquisition and triggers more competition for nutrition[50,51]. This is confirmed by the increasing volume of deciduous shrubs associated with lower graminoid and forb cover[52]. Therefore, higher woody taxa abundance during the Holocene reflects an environment dominated by competition, supporting the negative richness to range-size relationship.

In line with the relative change in cushion plants and trees, the Stress Gradient Theory[19] posits that positive interactions are more prevalent in stressful environments (glacial period), which contributes to taxa range-size expansions[6], whereas negative interactions are more common in favourable environments (Holocene period), constraining the range-sizes of the co-existing taxa. Overall, our study supports the plant interaction hypothesis, which builds upon the Stress Gradient Theory: during the glacial period, plant interactions were predominantly positive, which may contribute to a positive richness to range-size relationship, while during the Holocene period, negative interactions may result in a negative relationship. This study, however, is simplified by relying on the correlation of selected drivers to uncover causal evidence, and the incorporation of a model is essential, such as an individual-based model[53], a multispecies model[54], or a species distribution model[55]. Nevertheless, the implications of the shift in the richness to range-size relationship cannot be overemphasised.

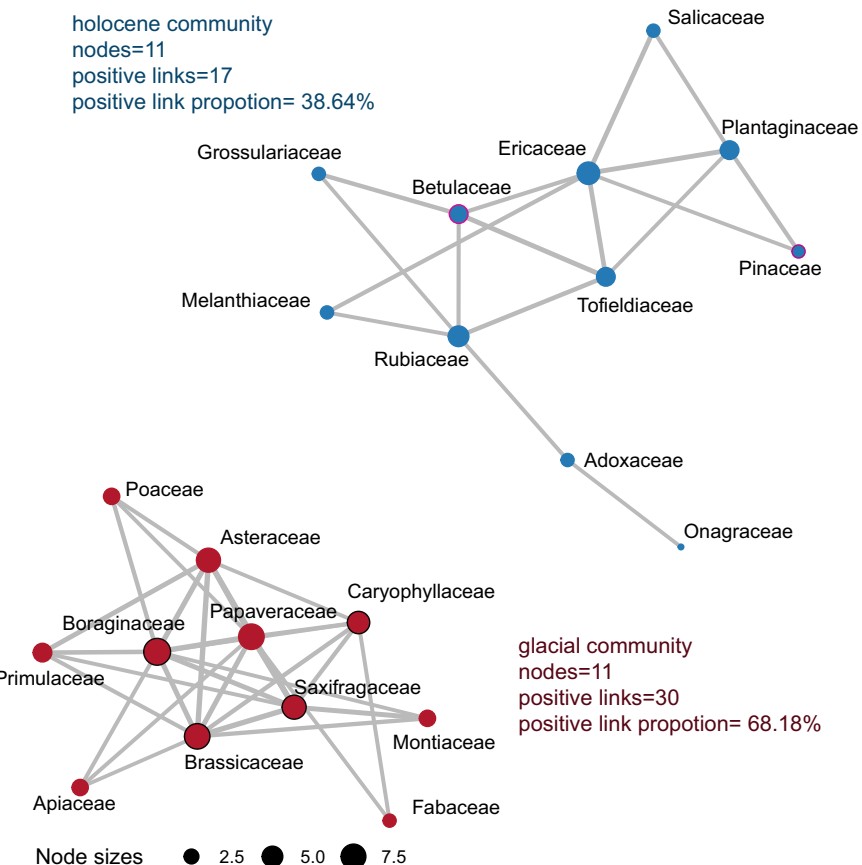

**Fig. 6 | Positive correlation network of plant groups.** Plant families (cushion plant taxa are outlined in black, including Boraginaceae, Caryophyllaceae, Saxifragaceae, and Brassicaceae; tree taxa are outlined in purple, including Betulaceae and Pinaceae) are represented by nodes, where the size of the node represents the number of links (node degree). Edges indicate positive correlations between plant taxa. The Holocene and glacial communities are shown separately. Source data are provided as a Source Data file.

## Conclusions and implications for plant conservation

Our study explores the plant richness to range-size relationship across time, which shifts from positive during the glacial period to negative during the interglacial period. Overall, our findings reject the hypotheses that either speciation or environmental heterogeneity are the main drivers shaping the richness to range-size relationship. Instead, we conclude from our investigation of sedaDNA data from lake sedimentary archives covering the last 30,000 years that positive plant interactions may result in a positive richness to range-size relationship while negative interactions yield a negative relationship.

In the future, to conserve tundra biodiversity effectively, attention should be towards the far northern tundra region. Within the context of climate change, colonisation in the present-day subarctic regions by trees and shrubs promotes the establishment of negative interactions, thereby constraining the range-size of the tundra taxa and hampering protection of the taxa. In the boreal region, certain understorey herb species with limited competitiveness will exhibit restricted range-sizes. Likewise, their ability to migrate northward with climate warming may be hindered due to their poorer competitive ability. In contrast, in far northern tundra areas dominated by harsh environments, positive interactions may expand the taxa ranges and enhance the protection of tundra taxa, for example, translocations of the endangered species. Additionally, careful measures should be taken to prevent the introduction of alien species, especially those mediated by human activity, as the positive interaction characteristic of these regions makes them more susceptible to alien taxa invasion.

## Methods

### The sedaDNA dataset

Sedimentary ancient DNA was extracted from seven cores collected from seven lakes in the northeast Siberia plus Alaska region (Fig. 2a, Supplementary Data 1, Supplementary Note 1). DNA extraction, amplification, and high throughput next-generation sequencing are described in detail in Supplementary Note 2. In short, DNA was extracted using the PowerMax® Soil DNA Isolation Kit (Mo Bio Laboratories, Inc., USA) from cleanly taken subsamples across each core, purified and amplified using the g and h primer targeting the P6 loop of chloroplast *trn*L (UAA) intron[56]. Purified DNA sequences were sent to Fasteris SA (Switzerland) for sequencing on the Illumina platform. All steps were performed in the palaeogenetic laboratories at the Alfred Wegener Institute, Helmholtz Centre for Polar and Marine Research, Potsdam, that strictly follow ancient DNA protocols.

The age of each subsample was inferred using a Bayesian age-depth model[57–63]. All samples from 30,000 years ago until today were used in this analysis. The assignment of sequences to taxa was done using OBITools software v3[64]. First, the full sequence was recovered from the partial forward and reverse reads using the *obi alignpairedend* program. Next, the sequence was assigned with the samples by the unique tag combination using the *obi ngsfilter* program. Afterwards, the reads were dereplicated into unique sequences by the *obi uniq* program, and PCR or sequencing errors were removed using *obi clean*. Finally, the *obi ecotag* program was used for taxonomic assignment. Only Amplicon Sequence Variants (ASVs) matching 100% with the customised "SibAla_2023" database[32] were used in the analysis. The data quality control was performed by evaluating the PCR replicability.

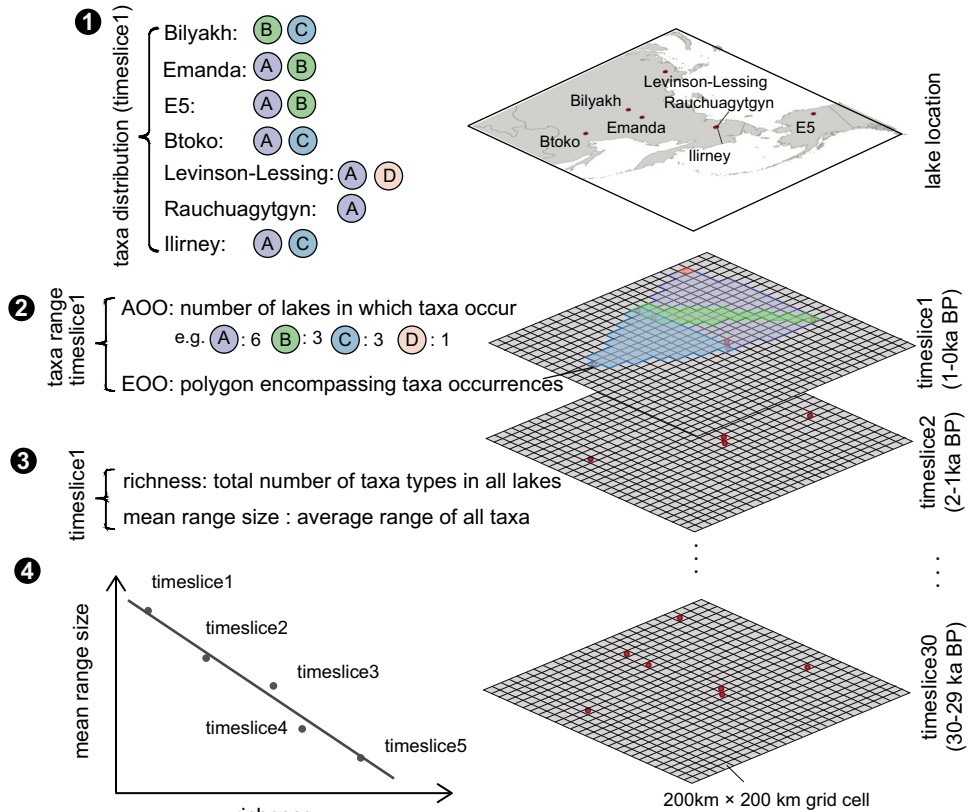

**Fig. 7 | Schematic diagram of the richness to range-size relationship calculation.** On the right, lakes are represented by a red dot, and remain the same for the past 30 time-slices, the map was created using the Free and Open Source QGIS. The study region was divided into 200 km x 200 km grid cells, which were used to evaluate the taxa ranges. On the left, A, B, C, and D represent different taxa that are shown in different coloured circles. Taking time-slice1 as an example, taxa distribution is available from the sedaDNA data (step 1). For the AOO (Area of Occupancy) method, the range of a specific taxon corresponds to the number of lakes in which the taxon occurred (step 2). For the EOO (Extent of Occurrence) method, the range was calculated as the sum of the grid cell areas overlapping with the convex hull spanning the lakes in which a specific taxon occurred (step 2). The richness in every time-slice is the total number of taxa types of all lakes, which is 4 in time-slice1; the mean range-size refers to the average range-size of all taxa (step 3). After computing the richness and mean range-size for each time-slice, a linear regression model was run to show the relationship between these variables within 5,000-year time-windows (step4).

At ASV level, the compositional data were transformed using the "Hellinger" method first (decostand() function in R package vegan[65]). Afterwards, the similarity of the three PCR samples was evaluated using non-metric multidimensional scaling (NMDS) (metaMDS() function in R package vegan[65]). In the case that the replicates of the same sample show a pronounced distance from all other samples, which indicates low-quality replicates, the PCR sample was excluded. In addition, if the replicates of the same sample did not form a cluster, distinct compositions among the replicates were assumed, and the corresponding PCR samples were excluded.

The "SibAla_2023" database was built with the following steps: 1. Taxa selection from a given region (55–90°N, 50–150°E and 40–90°N, 150°E–140°W) and taxa occurrences (>10) using the Global Biodiversity Information Facility (GBIF)[33] resulting in 233 families, 1059 genera, and 4849 species. 2. Alignment between selected taxa and available P6 loop sequences from public databases (arctborbryo[66–68]; EMBL 143[69]; PhyloNorway[30]). 3. Quality filtering of selected P6 loop sequences. 4. Preparation for usage with "obi ecotag". The SibAla_2023 database compared with given occurrences in GBIF amounts to 95.7% (family-level), 89.4% (genus-level), and 70.1% (species-level) of taxonomic coverage. Finally, the SibAla_2023 database includes a total of 4939 entries, comprising 3398 species, 947 genera, and 223 families that collapse into 2371 unique P6 loop sequence types.

For this study, we divided the above sedaDNA dataset into 30 time-slices, each spanning 1000 years. Since the number of matched ASVs (read counts) differs across lakes and time-slices, and plant richness estimates may increase with read counts resulting in potential bias, we performed a resampling analysis. For each lake and time-slice we rarefied the datasets to equal counts (base count = 5000) and repeated this process 100 times.

**Plant taxa richness**

Richness was defined as the total number of taxa types, indicated by the corresponding ASV types, per time-slice across all study lakes in the study region (Fig. 7).

**Mean range-size**

The mean range-size ("range-size" for simplicity) refers to the average distribution range-size of all taxa (represented by corresponding ASVs in this study) in every 1,000-year time-slice within the study region. We applied two established methods (AOO – Area of Occupancy, EOO – Extent of Occurrence) to calculate the range of taxa within our study region[70]. For AOO, the range of specific taxa corresponds to the number of lakes in which the taxa occurred. For EOO, we created grids with a cell size of 200 km x 200 km over the study region on an equal area projection (Lambert Azimuthal Equal Area with the study region as the centre). Next, the range-size was calculated as the sum of the grid-cell areas overlapping with the convex hull spanning the lakes in which a specific taxon (ASV) occurred. In the case of one lake, the range-size corresponds to the grid-cell area of the lake (Fig. 7).

We evaluated the method, i.e. the lake number in defining range-size, based on modern plant species occurrence derived from the Global

Biodiversity Information Facility (GBIF) database[33]. The plant species occurrence data were retrieved using the occ_search() function of the rgbif package[71,72] in R (no doi number was assigned), covering the regions of the sedimentary lake core, within the following coordinates: 55–76°N, 50–150°E, 54–76°N, 150–170°E, and 54–76°N, 180°W–140°W, on October 13, 2023. The data could be found in the source data. Using a uniform grid cell size of 200 ×200 km within the Siberian and Alaskan regions (Supplementary Note 4, Supplementary Fig. 5), we calculated the taxa range-size based on the AOO method. In each grid cell, richness was determined by the total number of taxa types present, and the mean range-size was calculated as the average range of all taxa. Spatially, richness and mean range-size across grid cells were used to fit a linear regression model using the lm() function from the R stats package[73] (Fig. 2c). To validate the taxa range-sizes using the seven lakes method, we assessed the consistency of range-sizes of modern plant taxa between the seven lakes region and the larger Siberia and Alaska region. We extracted taxa present in both the 200 km x 200 km grid cells centred on the seven lakes (seven grid cells) and those within the northeast Siberia and Alaska region. We then compared the range-sizes of the same taxa by summing occupied grid cells centred on the seven lakes and those within the northeast Siberia and Alaska region (Fig. 2d).

**The relationship of richness and mean range-size.** The richness to range-size relationship in this study refers to the relationship between richness and mean range-size. Based on the 100 interaction values for richness and mean range-size in each time-window (5 time-slices (Supplementary Note 4, Supplementary Fig. 4)), a linear regression model was fitted using the lm() function from R stats package[73] and is illustrated in Fig. 4. Subsequently, posterior simulations of sigma were obtained from the lm object using the sim() function from the arm package[74], and displayed in Fig. 3c.

### Environmental heterogeneity

To quantify environmental heterogeneity over time, we used beta diversity within the sedaDNA dataset as a proxy. Beta diversity represents the dissimilarities of multiple sites, which constitutes two components: richness differences (or nestedness indicating the richness difference between sites) and species replacement or turnover[75,76]. To derive beta diversity estimates independent from richness we used species turnover only to quantify heterogeneity. The calculation was done using the beta.multi function in the betapart R package with 'jaccard' as the family index[77].

### Biotic interactions

We used the relative abundance of trees as a proxy for an environment dominated by negative interactions[52], and relative cushion plant abundance as a proxy for environments with more positive interactions[43,44]. The relative abundance was calculated as the ratio between the number of cushion plant/tree taxa observations and the sum of all taxa observations within the time-slice. To test the plant interaction hypothesis, two generalised linear models were constructed, with the richness to range-size relationship as a binomial response variable (R > 0.2 for a positive relationship, and R < -0.2 for the negative relationship) and cushion plant abundance and tree abundance as explanatory variables (Fig. 5).

### Temperature reconstruction

We reconstructed the mean annual temperature over the last 30,000 years for the study region based on pollen assemblage records from ten lakes[23]. For each lake's pollen record, modern pollen assemblages from within a 1,000 km radius were compiled to form a calibration set, and used to model temperature downcore with the WAPLS function in the rioja R package[78]. In addition, the oxygen isotopic composition ($\delta^{18}O$) from the North Greenland Ice Core Project (NGRIP)[24] is used as a temperature index, and shown in Fig. 3h.

### Network analysis

All the terrestrial plant taxa were grouped to the family level: 37 plant groups were kept after applying a filter for a minimum of 10 read counts across all time-slices and occurrences in at least 5 time-slices. We then calculated the pairwise Spearman rank correlation of these groups using the rcorr() function from the Hmisc package in R[79]. Only correlation coefficients more than 0.6 (adjusted $p < 0.05$) were used to build an undirected positive interaction network with the igraph package[80,81]. Next, the network was clustered into two communities according to the cluster_optimal() function in the igraph package.

### Methods restriction and evaluation

The estimation of range-size from sedaDNA was based on continuous records using both the AOO method and the EOO method, resulting in highly consistent findings. However, this study exhibits several limitations. Firstly, AOO may result in an underestimation of range-size due to incomplete sampling (seven lakes from across the region), while EOO may overestimate range-size due to noncontinuous distributions or fragments[82]. Secondly, environmental heterogeneity might have been underestimated, as abiotic features such as topography and soil characteristics have not been included. Thirdly, the number of inter-actions might be overrepresented in network analyses due to indirect effects, wherein both taxa respond to another variable, such as the environment, which has not been removed. Finally, some drivers that influence the richness to range-size relationship might have been overlooked in this study, requiring further investigation.

All data analyses were done in R version 4.3.2[73].

### Reporting summary

Further information on research design is available in the Nature Portfolio Reporting Summary linked to this article.

## Data availability

The raw sedaDNA sequence data have been deposited in the European Nucleotide Archive (ENA) at EMBL-EBI under accession number PRJEB76237. Detailed information for every lake sedimentary core is available here: Lake Billyakh: https://doi.org/10.1016/j.quascirev.2010.04.024; Lake Bolshoe Toko: https://doi.org/10.3389/fevo.2021.625096; Lake E5: https://doi.org/10.1016/j.quascirev.2018.12.003; Lake Emanda: https://doi.org/10.1111/bor.12476; Lake Ilirney: https://doi.org/10.1016/j.quascirev.2020.106607; Lake Rauchuagytgyn: https://doi.org/10.5194/bg-18-4791-2021; Lake Levinson-Lessing: https://doi.org/10.1002/jqs.3384. The pollen data used for the temperature is available from Zenodo: https://doi.org/10.5281/zenodo.7887565. Source data are provided with this paper.

## Code availability

R scripts for processing the data, with dataset input provided with this paper, can be downloaded via Zenodo: https://doi.org/10.5281/zenodo.11259177.

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

## Acknowledgements
We thank Cathy Jenks at the University of Bergen for proofreading this manuscript. This research has been supported by the European Research Council (ERC Glacial Legacy 772852 to Ulrike Herzschuh), and China Scholarship Council (grant 202106620011 to Ying Liu).

## Author contributions
Y.L. and U.H. conceived the manuscript and designed the study, Y.L. was supervised by U.H., J.C. and K.R.S. created the database, assigned sequences, and conducted quality control. Y. L., U.H., and S.L. processed, analysed, and interpreted the data. Y. L., U.H., and S.L. drafted the manuscript. Y. L., U.H., S.L., J.C., and K.R.S. contributed to the manuscript revisions.

## Funding

## Competing interests
The authors declare no competing interests.
