## [Peer Review File · Nature Communications]

Plant interactions associated with a directional shift in the richness range size relationship during the Glacial Holocene transition in the ArcticREVIEWER COMMENTS

Reviewer #1 (Remarks to the Author):

Overall, I think the manuscript is well written – the authors present a rather convincing picture of the relationship between the species range and the taxonomic richness. The interesting pattern that the relationship is positive in the glacial times, and changes to a negative relationship that is at present observed in many areas is convincing and novel. The conclusion that this pattern is partly the result of species interactions seems strong, although future work will be required to see if this holds. The ancient DNA metabarcoding is utilized to the fullest and was done carefully - with an impressive nearly complete reference database. I think the paper would improve from an improved introduction to really clearly defining the three hypotheses – see specific comments. I understand the idea behind the hypotheses, but miss the logical connection between the hypotheses and the expected relationship. Especially for a general audience this needs to be simple and crystal clear. Further, the conclusion could do with a bit more emphasis on the implications of the results in a broader context e.g., conservation and possible human impact in this relationship.

Abstract

Line 14-15: I think the first sentence of the abstract is not really fully comprehensible. Mainly 'richness' and 'mean range size' are very vague and will not mean much to many readers. Think it has to be made clear that richness is related to taxonomy, and mean range size is the area a species occurs in. If this is not clear in the first sentence, it is hard to understand why there would be a relationship and you might lose some readers.

Introduction

Line 40-42: there is nearly no mention of human-mediated species range size changes in the results and discussion of this manuscript, as well as there is little on the consequences of these results in terms of conservation. Maybe it is worth elaborating a bit more in the conclusion on this. The human-mediated species range size is not tested at all, and maybe should either be discussed or removed here.

Line 50-51: Is this really true? which extinctions are meant here – I had a quick look at reference 12 (Alahuhta et al., 2020) does not mention extinctions, only risks. I assume these are local extinctions and not the extinction of a species in its entirety (unless megafauna is meant). This should be clarified. There are nearly no plant extinctions which the authors mentioned in lines 154-155.

Line 43-56 and line 95-98: I do not fully understand this argument. If this relation of range and species richness is the effect of temperature-dependent speciation/extinction, assuming that none of these occurred during the last 30 kyr (no speciation and no extinction), the negative relation that is observed today should also have been present in the past. I understand that the result of changing this relationship will not be relevant on human time scales. But shouldn't Figure 1c show a negative correlation that is stable? Every single time slice in the past 30 kyr should be similar.

Line 63-65: I have trouble understanding this last point, and how this is following from the environment heterogeneity hypothesis. Wouldn't human landscape homogenization remove any relationship since areas with rich flora but small ranges would not be there anymore? I see how homogenization would result in a large range and low richness, but I don't understand how the relationship can still be maintained. (e.g., the relation would remain negative if you consider the 30 ka together, but only looking at one time slice would not show a relationship?)

Line 94-95: I think calling this study design 'straightforward' is understating the underlying complexity of the study – I would remove this.

Line 98-100: I don't follow completely the reasoning behind a consistent relationship, if hypothesis 2

is true, wouldn't heterogeneity be co-correlated with changes in the richness-mean range size relationship in the study area? If an increase or decrease in heterogeneity is not changing the relationship, this hypothesis can't be true. I think the introduction needs a bit more clarity on this hypothesis, and why a stable negative correlation is expected if this hypothesis is true.

Results

Line 112: I think it would be nice to have a percentage on the completion of the database, nearly 100% family coverage, and 90% genus level (taken from line 262-263) – this is incredible and really believe it should be emphasized, this directly puts more confidence in the results of this manuscript – I personally really appreciate these numbers!

Line 130-141: The order of the panels in Figure 4 is confusing to me and would like to see the panels in order as they are referred to in the text. Panel (a) is the range/richness relationship, and I understand this is the most important result, reading the text starts by referring to panel (b).

- I think panel (a) should be moved down. In the text, (b) and (c) are mentioned first. It makes more sense that the relationship between richness and mean range size should come after showing the individual values.
- Panel (a) has a colour gradient that is not explained in the caption – either explain or remove the gradient.
- Maybe it's possible to put some kind of separator after panel (c) to make clear that panels b and c make a, but that the other plots are independent.

Line 155-157: Is the fact that climate warming supports immigration northward relevant to this hypothesis? Is the hypothesis more inclusive and sees speciation as the appearance of a novel taxa in an area, and extinction as the local disappearance of a taxa? – something that can happen at millennial time scales? I think this sentence is blurring the definition of the hypothesis.

Conclusion:

Line 221-235 I miss a little bit more on the implications of the results, and what the impact for conservation is, as well as the potential human impact of this relationship.

Minor edits:

Line 34: Maybe add 'status' after 'conservation'

Line 114-117: Taxa and ASVs are used interchangeably here – as well as in lines 271-272. Maybe for a general audience, it is better to call them plant taxa and mention in the methods that the taxa are derived from ASVs.

Line 117-118: remove the brackets, and write that both; Area of Occupancy (AOO) and Extent of Occurrence (EOO) were done, but that results were similar and EOO was presented in the text, while AOO is shown in the supporting information. The sentence in the brackets is a bit confusing and requires you to check the supporting information to understand what's going on. With one clear sentence, the paragraph flows better.

Line 135, it would be nice to have a time range for the early Holocene, and for the second maximum – similar to what was done in line 133. Makes it easier to quickly find the right data point in the figure.

Line 140: Is the relationship truly ubiquitous? The introduction says nearly – I think this statement needs to be lowered a bit here.

Line 147: Why "also"

Line 212: Maybe worth changing 'benign' to a simpler word, I had to look it up.

Line 216-217: maybe remove "well-thought-out design", it should be assumed that the study was designed well – no need to write this down.

Line 256: remove "basically"

Figure 4 is cited in the text before Figure 3.

Reviewer #2 (Remarks to the Author):

This paper explores the temporal relationship between richness and mean range-size in northeast Siberia plus Alaska. The authors use plant sedaDNA and the plant local DNA "SibAla_2023" database to reconstruct past plant richness and distribution changes over the last 30,000 yr in seven lake sediment records at a millennial resolution. They propose three hypotheses to explain the widely observed negative richness to mean range-size relationship. Additionally, the authors use a mean annual temperature inferred from ten pollen sites in the study region.

Overall, I found the manuscript really interesting and well written. I believe the paper will make a valuable contribution to Nature Communications and significantly impact conservation efforts for northern plant species. However, I did encounter some challenges in following the explanation of the three hypotheses. Shortening the sentences for clarity and providing a brief explanation of mean range-size would enhance readability. Below, I summarize my minor comments and provide line-by-line suggestions.

Minor comments:

- Regarding "range-size," it would be beneficial to standardize its usage, either with or without hyphens throughout the manuscript.
- Sentence 163-165 requires rephrasing for clarity.
- For sentence 138, an explanation of the lack of significance for the time slice 13-9 ka would be helpful.
- The sentence at 230-232 is difficult to read and should be rephrased.
- For sentence 254-255, The authors reference a paper for the data quality control that is on reviewing process. It would be helpful to include some relevant information in this paper if feasible.

Figures:

- Fig. 2: Consider changing dot colours for each lake in Figure 2c to improve differentiation.
- Fig. 4: If Figure 4 appears in the text before Figure 3, it would be advisable to switch their numbering.
- Fig. 3: Diverging colours could enhance the clarity of range-size in Figure 3.
- Fig. 4: Adding a Northern Hemisphere temperature reconstruction not inferred from vegetation to Figure 4g for comparison with pollen-based regional temperature would be beneficial.

Reviewer #3 (Remarks to the Author):

In this research, the authors used sedimentary ancient DNA in cores across Siberia and Alaska to examine how the relationship between plant richness and range size changes through time. In the present day, this relationship is nearly ubiquitously negative (sites with higher species richness have smaller average range sizes); however, the authors found that this negative pattern has not persisted through time, linking changes in the richness-range size relationship to positive interactions/facilitation by cushion plants and negative interactions/exclusion by larger tree species. The results of this research (that the richness-range size relationship has not always been negative) are very interesting and could provide exciting new insight into how communities are structured across space and time. However, I have a few concerns I'd like to see addressed before I can fully agree with the mechanism/hypothesis put forward to explain this temporal variance.

First, I believe a moderate expansion on the methods is necessary to ensure replicability. The data collection and management are both very well documented, but, aside from the network analysis, the

analyses conducted on the richness-range size relationship need a bit more information. In particular, it was unclear how the actual relationship was calculated (I assume a linear model of some sort, and correlations were mentioned in line 137, but I'm unsure how the uncertainty estimates were generated), and why the sliding window analysis (not mentioned until Lines 137) was necessary. Similarly, although the GBIF analyses is mentioned in the methods, it was unclear in the remainder of the paper (and especially in Figure 2b) what was done. Expanding and clarifying the methods would go a long way to make the research easier to understand and interpret.

Second, I very much appreciate the inclusion of the three hypotheses for the different relationships expected, and believe these are necessary for the interpretation of the paper. However, the justification behind the expected patterns based on the three hypotheses is often unclear. I think that my confusion might stem from the fact that the manuscript bounces back and forth between dealing with temporal trends (i.e., the time series patterns of the richness-range size relationship) and spatial trends (i.e., whether the range size-richness patterns are positive or negative at a single time point). I expand on the places I was unclear about in my line-level comments.

Abstract:

Line 15: Unclear what is meant by "inherent temporal dimensions"

Line 17: I would just say "richness-range size relationship"; "richness-mean range-size" sounds like you're taking the mean of richnesses instead of the range sizes. Alternately, "the relationship between richness and mean range size" as before.

Line 18: the major factors of what? The identity of the richness-range size relationship or its change through time? This difference drives most of my confusion with regards to the interpretation of the hypotheses. See my comments on lines 53-56 too.

Line 24: see comments on line 18

Introduction:

Line 35: either "nearly ubiquitous across space" or "nearly ubiquitously observed across space"

Line 39: But mean range-size itself doesn't expand, it's often the fact that smaller-ranged species are going extinct/reducing their ranges that leads to an increase in mean range size at a site.

Line 45: "106 and 105 years" is a perfectly fine way to write this, but maybe "thousands to millions of years" would be clearer? Definitely don't feel obligated to change it, though, it's fine as it is.

Line 46-53: The example provided here (the latitudinal diversity gradient) occurs on a global scale: what about the same relationships on local or regional scales, where there are still differences in diversity from speciation/extinction rates? That would get at the next hypothesis (heterogeneity), but because these processes act on very different spatial scales in addition to temporal scales, I'm not sure they're directly comparable. Maybe the differences in scale are the point, though – is this research examining the differences in spatial and temporal scale of these drivers explicitly?

Lines 53-56: These statements are unclear for a few reasons: First, the main points of the paper deal with changes back through time, but this statement talks about future dynamics and human-relevant timescales, which are not really examined in the paper. In addition, is there reason to believe that species extinctions (especially in a regional area like Siberia+Alaska) cannot happen on the timescales studied? Finally, the justification for the speciation hypothesis is entirely spatial (i.e., areas of higher vs. lower species richness), so I'm confused where the interpretations for temporal changes in this relationship come from (see comments on line 18).

Also, if speciation rates happen faster when it is warmer and extinction rates are slower (as mentioned in lines 46-47), causing a more strongly negative richness-range size relationship, wouldn't you expect negative relationships in the Holocene and less negative relationships in the glacial period (exactly what is found in the results)?

I just think a little bit more explanation would really help the interpretation of this hypothesis.

Line 57: Very well-reasoned and good justification! As mentioned before, environmental heterogeneity is a smaller scale (spatial and temporal) than the previous hypothesis, but maybe that's the point?

Line 68: I might rearrange this sentence to ensure that the stress-gradient hypothesis is defined before talking about the hypothesis on how it could be applied to the case study of this manuscript. Here's one possible suggestion: "The stress-gradient hypothesis (cite) suggests that in less stressful environments, high richness can _____. In contrast, _____", and then bringing up how this hypothesis might be applied to temporal change.

Line 75-77: This hypothesis assumes that the species are less stressed during times of warming, which may well be the case, but some very cold-adapted species (especially in the lower latitudes) might be more stressed in warmer temperatures than cold ones.

Line 80-86: This is very clear and understandable!

Line 88-91: You may want to split this sentence into two (maybe at the ", which,")

Line 92: change "exploit" to "use"

Line 95-98: See my comments on lines 18, 53-56: I think it is possible to see a significant relationship that changes through time just based on small-ranged species going extinct.

Line 99: "consistently" instead of "stably"
Results and Discussion:

Lines 108-118: Very well explained!

Line 122: Although Figure 2 alludes to it, I believe this is the first mention of the GBIF analyses. Moving the subsequent sentence (starting with "Furthermore, the range size of GBIF-derived plants") before this sentence would help a lot in clarifying what the GBIF analyses are. Maybe even the sentences in the Methods about this (Lines 281-285) would be helpful here.

Line 126-129: Cool! And good to know that the method of calculation doesn't appear to affect your results!

Line 137: Why do 5-ka time windows for the sliding-window analysis? How does the breadth of this window affect your analyses (e.g., 5ka vs 2ka vs 10ka windows)?

Line 137: revise to "for the glacial period (30-11 ka) and negative..."

Line 138: Were the p-values in Supplementary Table 2 (and shown in Figure 3) corrected for multiple comparisons? If not, I'd suggest using the `p.adjust()` function in `r` to do so.

Lines 146-148: See comments on line 18, Lines 53-56

Lines 175-176: What does a positive vs. negative relationship mean in this case? Effect size > 0 vs. effect size < 0 ? I think you'd want to define a threshold (say, $R > 0.2$ or < -0.2), because a slope of,

say 0.000001 is not particularly biologically significant.

Line 177: This is the first time that cushion plants are mentioned – I think parts of the excellent explanation provided in Lines 188-208 should be moved before this paragraph to prime the reader for the ecological role of cushion plants and why they are important for the study.

Line 182-185: Very interesting, I love the idea of using positive network interactions! I know this is mentioned in the methods, but maybe make it clear that the internal edges are all positive interactions. Also, I'd be interested to know about the proportion of positive interactions, in addition to the absolute number of them.

Lines 216-219: Revise to something more like "However, this study relies on the correlation of selected drivers. To uncover causal evidence, ..." Also, "the incorporation of a model" is somewhat vague, I think it could be improved by adding one or two specific examples of what can be done to uncover causal evidence.

Line 228: Maybe revise from "In analogy to the past" to "In the future" or something like that, where the future implications are highlighted as opposed to the past (which is what the main body of the paper discusses).

Methods:

Lines 273-280: why 200 by 200 km grid cells? Does using different spatial resolutions change the results?

Lines 286-293: Needs more citations (e.g., Baselga et al. 2010 GEB for the sentence ending on line 291). Also, cite the r package used.

Lines 287-288: some methods of beta diversity are calculated like this, but the way you're calculating beta diversity (with Jaccard turnover) is not.

Line 291: Very cool, I like the fact that you're accounting for the richness differences by using turnover specifically instead of both turnover and nestedness.

Lines 294-298: Would it be possible to compare the proportion of cushion/tree taxa to the number/proportion of positive interactions directly? It appears from Figure 6 and from the results that they constitute two different communities, one with many more positive interactions than the other, but I would like to see that explicitly stated in the paper.

Also, to clarify: The relative abundance was calculated by the number of taxa (i.e., the diversity) of cushion plants/trees, or the number of individual observations? If it's the number of taxa, I'm not sure that gets at dominance, just diversity.

Line 299-303: Using the pollen assemblages to reconstruct the temperature is going to be circular, given that the study examines changes in plant assemblages through time. This isn't to say you should remove it; instead, I'd suggest comparing both the pollen-reconstructed temperature and temperature derived from a different method (e.g., global circulation models) and using both in the manuscript and in Figure 4.

Line 324: List the packages used (igraph, betapart, others) with citations either here or in a supplement.

Figure 1: I appreciate the conceptual figure of the different hypotheses and how they might look along the landscape. However, I think this figure falls into the same trap of looking at temporal patterns vs. spatial patterns: For example, if I am reading the hypotheses correctly from the introduction, the speciation hypothesis would cause spatial patterns in richness/range size, but not temporal patterns.

Figure 1c appears to show that there would be no relationship whatsoever if the speciation hypothesis holds. The other two hypotheses are very well-described in the figure, though! Maybe adding a separate column of figures like 1c, 1e, 1g, and 1i looking at the hypothesized relationships in the interglacial period and change the others to the hypothesized relationships in the glacial period specifically would help to clarify.

Figure 2:

Figure 2a: I don't think the coordinates along the edge of map box are necessary with the projection: because it's close to polar, the latitude and longitude lines are not perpendicular to the edges of the map box or to each other. You could add longitude lines in the map if you'd like, but I don't think the map needs it.

Also, (and this applies to Figure 2c as well), I would color-code the different sites so it's clear where each site is and the richness and mean range size of each. I only see 6 out of the 7 sites -- I assume Rauchaogytgyn and Ilirney are very close together, so I would jitter the map a little bit so that the reader can clearly see both.

Figure 2b: This is modern, correct? I would make it very clear that it is modern range sizes.

Figure 2c: Color code the different sites. This will also help to distinguish this plot from Figure 2d, which is (I believe) modern occurrence data.

Figure 2d: This is modern, correct? I would make it very clear that it is modern range sizes.

Caption: instead of saying "(by counting the number of grid cells in which each taxon occurs)", you can say "...range size of selected taxa based on the sum of occupied 200x200 km grid cells"

Figure 3:

This is a beautiful figure!

Caption:

Line 528: I would add how the average range size is calculated (mean number of lakes occupied, right?)

Line 528: Typo "coloured"

Figure 4: Good! The subheadings of the figures might be better outside the boxes, because at the moment they are a little hard to see

Figure 5: This is a very clear figure and it really shows the importance of cushion plants/tree proportion in driving this relationship. See my comments on Line 175-176.

Figure 6:

Good! Maybe use different colors for the two communities, because they were previously used in Figures 3,4 for cushion plants vs. trees. Also, maybe distinguish the tree and cushion plant taxa on the graphs themselves (e.g., putting a black border around the cushion plant nodes). Finally, I think the overall graphs (including both positive and negative associations) should be in the supplements.

Response Letter

All changes were marked using the “**Track Changes**” function in the revised manuscript.

Reviewer #1 (Remarks to the Author):

Overall, I think the manuscript is well written – the authors present a rather convincing picture of the relationship between the species range and the taxonomic richness. The interesting pattern that the relationship is positive in the glacial times, and changes to a negative relationship that is at present observed in many areas is convincing and novel. The conclusion that this pattern is partly the result of species interactions seems strong, although future work will be required to see if this holds. The ancient DNA metabarcoding is utilized to the fullest and was done carefully - with an impressive nearly complete reference database.

I think the paper would improve from an improved introduction to really clearly defining the three hypotheses – see specific comments. I understand the idea behind the hypotheses, but miss the logical connection between the hypotheses and the expected relationship. Especially for a general audience this needs to be simple and crystal clear. Further, the conclusion could do with a bit more emphasis on the implications of the results in a broader context e.g., conservation and possible human impact in this relationship.

Response: We are thankful to Reviewer #1 for their constructive comments.

Abstract

Reviewer comment: Line 14-15: I think the first sentence of the abstract is not really fully comprehensible. Mainly ‘richness’ and ‘mean range size’ are very vague and will not mean much to many readers. Think it has to be made clear that richness is related to taxonomy, and mean range size is the area a species occurs in. If this is not clear in the first sentence, it is hard to understand why there would be a relationship and you might lose some readers.

Response: agree, revised.

New text: Line 14-15: A nearly ubiquitous negative relationship between taxonomic richness and the mean range-size (average area of the taxa occurrence) is observed across space.

Reviewer comment: Line 40-42: there is nearly no mention of human-mediated species range size changes in the results and discussion of this manuscript, as well as there is little on the consequences of these results in terms of conservation. Maybe

it is worth elaborating a bit more in the conclusion on this. The human-mediated species range size is not tested at all, and maybe should either be discussed or removed here.

Response: agree, we removed the “human-mediated species range size” here. In the conclusion and implications section we include the human-mediated information.

New text: Line 43-44: Alternatively, do changes in species range-size result in richness adjustments?

New text: Line 286-288: Additionally, careful measures should be taken to prevent the introduction of alien species, especially those mediated by human activity, as the positive interaction characteristic of these regions makes them more susceptible to alien taxa invasion.

Reviewer comment: Line 50-51: Is this really true? which extinctions are mend here – I had a quick look at reference 12 (Alahuhta et al., 2020) does not mention extinctions, only risks. I assume these are local extinctions and not the extinction of a species in its entirety (unless megafauna is meant). This should be clarified. There are nearly no plant extinctions which the authors mentioned in lines 154-155.

Response: Agree, we change the extinction to “regional species extinctions risk”. And we change the sentence about “no plant extinctions”.

New text: Line 55-57: In contrast, low temperatures and recurrent extensive glaciation in high-latitude regions have resulted in less speciation and increased the risk of regional species extinctions.

New text: Line 192-194: However, the period of our study is too short to accumulate sufficient mutations for speciation which requires millions of years, and the effect of extinction was eliminated by using taxa with 100% assignment to the database.

Reviewer comment: Line 43-56 and line 95-98: I do not fully understand this argument. If this relation of range and species richness is the effect of temperature-dependent speciation/extinction, assuming that none of these occurred during the last 30 kyr (no speciation and no extinction), the negative relation that is observed today should also have been present in the past. I understand that the result of changing this relationship will not be relevant on human time scales. But shouldn't Figure 1c show a negative correlation that is stable? Every single time slice in the past 30 kyr should be similar.

Response: We are sorry for the misunderstanding; the spatial scale study tests the relationship across different sites, as shown in Figure 2b,c. We tested the relationship of richness to mean range-size over a temporal scale (across timeslices with a 5,000-year time window approach) rather than spatial scale. A schematic plot has been added to show the calculation process more clearly (Figure7). We revised Figure 1 to show the logical connection between the hypotheses and the expected relationship.

New Figure: Line705-709:

Figure 7: Schematic diagram of the richness to range-size relationship calculation. On the right, lakes are represented by a red dot, and remain the same for the past 30 timeslices. The study region was divided into 200 km x 200 km grid cells, which were used to evaluate the taxa ranges. On the left, A, B, C, and D represent different taxa that are shown in different coloured circles. Taking timeslice1 as an example, taxa distribution is available from the sedaDNA data (**step 1**). For the AOO (Area of Occupancy) method, the range of a specific taxon corresponds to the number of lakes in which the taxon occurred (**step 2**). For the EOO (Extent of Occurrence) method, the range was calculated as the sum of the grid cell areas overlapping with the convex hull spanning the lakes in which a specific taxon occurred (**step 2**). The richness in every timeslice is the total number of taxa types of all lakes, which is 4 in timeslice1; the mean range-size refers to the average range-size of all taxa (**step 3**). After computing the richness and mean range-size for each timeslice, a linear regression model was run to show the relationship between these variables within 5,000-year time-windows (**step4**).

New Figure: Line 617-638:

Figure 1: Evaluating three hypotheses of how species richness relates to range size over space (previous study) and time (this study), using a hypothetical richness increase scenario. Coloured polygons represent the distribution range of different plant taxa. Under the speciation hypothesis, regions experiencing high speciation rates over millions of years generate taxa with a constrained range, resulting in a negative richness to range-size relationship (a), in the temporal scale, no speciation occurs on millennial time scales, neither during the glacial period nor the interglacial period. The richness increases are not driven by speciation, and range-size remains stable or changes without a discernible pattern, thus, there is no richness to range-size relationship (b). Under the environmental heterogeneity hypothesis, the temporal scale pattern (d) mirrors the spatial scale pattern (c). Spanning both glacial and interglacial periods, heterogeneous environments favour a high richness of taxa with narrow ecological niches and limited geographic ranges, leading to a stable negative richness to range-size relationship (d). Under the plant interaction hypothesis, widespread negative richness to range-size relationships are observed on a spatial scale, attributed to negative interactions (e). Unlike the spatial scale result, in the temporal scale, a positive interaction during the glacial period expands taxa range-sizes (f), resulting in a positive

richness to range-size relationship; while a negative interaction during the interglacial period constrains the range-sizes of the taxa, resulting in a negative relationship. The relationship shifts from positive during the glacial towards negative during the interglacial (f).

New text: Line 48-67: First, long-term (millions of years) temperature-related speciation and species extinction may result in negative richness to mean range-size relationships at the spatial scale^{7,8}. Higher temperatures increase metabolic rates and promote speciation rates⁹. Within a high speciation region, new species are initially constrained within a small range, resulting in regions with high species richness and smaller range-sizes^{8,10} as exemplified by the equatorial region¹¹. In contrast, low temperatures and recurrent extensive glaciation in high-latitude regions have resulted in less speciation and increased the risk of regional species extinctions¹². Thus, speciation and extinction progress along the temperature gradient results in a negative richness to range-size relationship in space (Figure 1a).

However, this hypothesis applies to the global pattern, and is not solely attributed to temperature differences across space but also to the accumulation of the difference over millions of years.

Within a specific region and millennia scale, the impact of speciation and extinction may be limited. Hence, to shift the perspective to a temporal scale over millennia, richness and range changes are not driven by speciation. Moreover, without considering extinction, there is no expected relationship between richness and range-size across time (Figure 1b)

New text: Line 113-115: If million years of speciation and species loss are the primary factors, when applying this pattern to millennia studies, especially after eliminating the potential impact of extinction, no richness to range-size relationship would be expected

New text: Line 181-184: Given that we detect significant richness to range-size relationships for almost all time-windows in our study over the last 30,000 year, and that the potential extinction effect was eliminated by using taxa with 100% assignment to the database, we conclude that processes operating on millennial time-scales also influence the relationship.

Reviewer comment: Line 112: I think it would be nice to have a percentage on the completion of the database, nearly 100% family coverage, and 90% genus level (taken from line 262-263) – this is incredible and really believe it should be emphasized, this directly puts more confidence in the results of this manuscript – I personally really appreciate these numbers!

Response: agree, done.

New text: Line 133-135: The database has a taxonomic coverage of 95.7% at the family level, 89.4% at the genus level, and 70.1% at the species level, compared to occurrences in the Global Biodiversity Information Facility (GBIF) ³⁰.

Reviewer comment: Line 130-141: The order of the panels in Figure 4 is confusing to me and would like to see the panels in order as they are referred to in the text. Panel (a) is the range/richness relationship, and I understand this is the most important result, reading the text starts by referring to panel (b).

- I think panel (a) should be moved down. In the text, (b) and (c) are mentioned first. It makes more sense that the relationship between richness and mean range size should come after showing the individual values.
- Panel (a) has a colour gradient that is not explained in the caption – either explain or remove the gradient.
- Maybe it's possible to put some kind of separator after panel (c) to make clear that panels b and c make a, but that the other plots are independent.

Response: agree, revised.

New Figure: Line 661-682:

Figure 3: Changes in the plant richness to range-size relationship and potential drivers of the relationship. a, plant richness change over time; **b**, range-size change over time; **c**, the plant richness to range-size relationship shifts from positive to negative at the transition

from the glacial to interglacial (the relationship was calculated based on the richness and range-size, colour gradient is similar to that in Figure 4); **d**, biotic environmental heterogeneity over time; **e**, cushion plant abundance (%) over time based on the mean value of 100 resampling results of our sedimentary ancient DNA dataset from seven lakes in northeast Siberia and Alaska; **f**, tree plant abundance (%) over time based on the mean value of 100 resampling results of our sedimentary ancient DNA dataset from seven lakes in northeast Siberia and Alaska; **g**, reconstructed mean annual temperature (°C, with 95% confidence intervals) based on ten pollen sites in the study region using the weighted averaging partial least squares (WAPLS) method, over the past 30,000 year. Some sites are the same as the sedaDNA data sites. **h**, temperature index represented by oxygen isotope values ($\delta^{18}\text{O}$) over the past 30,000 years from the North Greenland Ice Core Project (GRIP)⁷⁰, higher values correspond to higher temperatures. To compare with the richness to range-size relationship (calculated based on 5,000-year (5 ka) intervals for each time window), and to smooth the data to eliminate noise, all the data are shown in 5,000-year running time windows.

Reviewer comment: Line 155-157: Is the fact that climate warming supports immigration northward relevant to this hypothesis? Is the hypothesis more inclusive and sees speciation as the appearance of a novel taxa in an area, and extinction as the local disappearance of a taxa? – something that can happen at millennial time scales? I think this sentence is blurring the definition of the hypothesis.

Response: We are sorry for the misunderstanding, climate warming supporting immigration northward is not relevant to this hypothesis, we removed the text.

Reviewer comment: Line 221-235 I miss a little bit more on the implications of the results, and what the impact for conservation is, as well as the potential human impact of this relationship.

Response: agree, revised.

New text: Line 277-288: In the future, to conserve tundra biodiversity effectively, attention should be towards the far northern tundra region. Within the context of climate change, colonisation in the present-day subarctic regions by trees and shrubs promotes the establishment of negative interactions, thereby constraining the range-size of the tundra taxa and hampering protection of the taxa. In the boreal region, certain understorey herb species with limited competitiveness will exhibit restricted range-sizes. Likewise, their ability to migrate northward with climate warming may be hindered due to their poorer competitive ability. In contrast, in far northern tundra areas dominated by harsh environments, positive interactions may expand the taxa ranges and enhance the protection of tundra taxa, for example, translocations of the endangered species. Additionally, careful measures should be taken to prevent the introduction of alien species, especially those mediated by human activity, as the positive interaction characteristic of these regions makes them more susceptible to alien taxa invasion.

Minor edits:

Reviewer comment: Line 34: Maybe add 'status' after 'conservation'

Response: agree, revised.

New text: Line 37: and can be regarded as an indicator for species conservation status.

Reviewer comment: Line 114-117: Taxa and ASVs are used interchangeably here – as well as in lines 271-272. Maybe for a general audience, it is better to call them plant taxa and mention in the methods that the taxa are derived from ASVs.

Response: agree, revised.

New text: Line 137-139: Total richness was calculated by counting taxa from all records for 1,000-year time-slices for the last 30,000 years and averaging over 5,000-year moving time-windows.

New text: Line 340-342: Plant taxa richness. Richness was defined as the total number of taxa types, indicated by the corresponding ASV types, per time slice across all study lakes in the study region (Figure 7).

Reviewer comment: Line 117-118: remove the brackets, and write that both; Area of Occupancy (AOO) and Extent of Occurrence (EOO) were done, but that results were similar and EOO was presented in the text, while AOO is shown in the supporting information. The sentence in the brackets is a bit confusing and requires you to check the supporting information to understand what's going on. With one clear sentence, the paragraph flows better.

Response: agree, revised.

New text: Line 141-145: Range-size was calculated from lake-wise resampled data by counting the lake number of each taxon's occurrence (AOO: Area of Occupancy method), and the area polygon covering all the lakes of taxa occurrence (EOO: Extent of Occurrence), respectively. The findings from both approaches exhibited concordance: the AOO result is shown in the text, while the EOO is shown in Supplementary Figure 1.

Reviewer comment: Line 135: it would be nice to have a time range for the early Holocene, and for the second maximum – similar to what was done in line 133. Makes it easier to quickly find the right data point in the figure.

Response: thank you, revised.

New text: Line 168-170: a relatively low value during the early Holocene (14-8 ka), and increases until the late Holocene (6-2 ka) with a second maximum (Figure 3b).

Reviewer comment: Line 140: Is the relationship truly ubiquitous? The introduction says nearly – I think this statement needs to be lowered a bit here.

Response: agree, revised.

New text: Line 176-178: Our results fundamentally differ from the widespread negative richness to range-size relationships previously reported at the spatial scale^{3,31}.

Reviewer comment: Line 147: Why “also”

Response: to express that at the millennial scale, there are other drivers that can influence the relationship of richness to range-size.

New text: Line 181-184: Given that we detect significant richness to range-size relationships for almost all time-windows in our study over the last 30,000 year, and that the potential extinction effect was eliminated by using taxa with 100% assignment to the database, we conclude that processes operating on millennial time-scales also influence the relationship.

Reviewer comment: Line 212: Maybe worth changing ‘benign’ to a simpler word, I had to look it up.

Response: thanks, revised.

New text: Line 258-259: negative interactions are more common in favourable environments (Holocene period).

Reviewer comment: Line 216-217: maybe remove “well-thought-out design”, it should be assumed that the study was designed well – no need to write this down.

Response: done.

Reviewer comment: Line 256: remove “basically”

Response: done.

Reviewer comment: Figure 4 is cited in the text before Figure 3.

Response: We have adjusted the order of Figures 3 and 4.

Reviewer #2 (Remarks to the Author):

This paper explores the temporal relationship between richness and mean range-size in northeast Siberia plus Alaska. The authors use plant sedaDNA and the plant local DNA "SibAla_2023" database to reconstruct past plant richness and distribution changes over the last 30,000 yr in seven lake sediment records at a millennial resolution. They propose three hypotheses to explain the widely observed negative richness to mean range-size relationship. Additionally, the authors use a mean annual temperature inferred from ten pollen sites in the study region. Overall, I found the manuscript really interesting and well written. I believe the paper will make a valuable contribution to Nature Communications and significantly impact conservation efforts for northern plant species. However, I did encounter some challenges in following the explanation of the three hypotheses. Shortening the sentences for clarity and providing a brief explanation of mean range-size would enhance readability. Below, I summarize my minor comments and provide line-by-line suggestions.

Response: We thank reviewer #2 for their constructive comments.

Minor comments:

Reviewer comment: Regarding "range-size," it would be beneficial to standardize its usage, either with or without hyphens throughout the manuscript.

Response: done, we standardize it to "range-size".

Reviewer comment: Sentence 163-165 requires rephrasing for clarity.

Response: revised.

New text: Line 201-206: We find low heterogeneity during the late marine isotope stage (MIS) 3 (30–24 ka) when using plant community beta-diversity between records as a proxy (Figure 4d). This confirms previous findings that taxa of the Eurasian mammoth steppe have a widespread distribution³⁵. While at the site scale this period may have the highest richness compared to other periods²⁸, at the regional scale, the low heterogeneity supports relatively low overall richness.

Reviewer comment: For sentence 138, an explanation of the lack of significance for the time slice 13-9 ka would be helpful.

Response: agree, revised.

New text: Line 173-176: All time-window correlations are significant ($p < 0.001$), except for the time slice 13–9 ka (see Supplementary Table 2 for details), which possibly reflects the overlapping signals of the glacial and Holocene periods.

Reviewer comment: The sentence at 230-232 is difficult to read and should be rephrased.

Response: agree, revised.

New text: Line 277-288: In the future, to conserve tundra biodiversity effectively, attention should be towards the far northern tundra region. Within the context of climate change, colonisation in the present-day subarctic regions by trees and shrubs promotes the establishment of negative interactions, thereby constraining the range-size of the tundra taxa and hampering protection of the taxa. In the boreal region, certain understorey herb species with limited competitiveness will exhibit restricted range-sizes. Likewise, their ability to migrate northward with climate warming may be hindered due to their poorer competitive ability. In contrast, in far northern tundra areas dominated by harsh environments, positive interactions may expand the taxa ranges and enhance the protection of tundra taxa, for example, translocations of the endangered species. Additionally, careful measures should be taken to prevent the introduction of alien species, especially those mediated by human activity, as the positive interaction characteristic of these regions makes them more susceptible to alien taxa invasion.

Reviewer comment: For sentences 254-255, The authors reference a paper for the data quality control that is on reviewing process. It would be helpful to include some relevant information in this paper if feasible.

Response: agree, revised.

New text: Line 315-323: The data quality control was performed by evaluating the PCR replicability. At ASV level, the compositional data were transformed using the “Hellinger” method first (decostand() function in R package vegan⁵⁶). Afterwards, the similarity of the three PCR samples was evaluated using non-metric multidimensional scaling (NMDS) (metaMDS() function in R package vegan⁵⁶). In the case that the replicates of the same sample show a pronounced distance from all other samples, which indicates low-quality replicates, the PCR sample was excluded. In addition, if the replicates of the same sample did not form a cluster, distinct compositions among the replicates were assumed, and the corresponding PCR samples were excluded.

Figures:

Reviewer comment: Fig. 2: Consider changing dot colours for each lake in Figure 2c to improve differentiation.

Response: done.

New figure: Line 640-653:

Figure 2: Plant richness to range-size relationship across space. **a**, Location of the seven lakes in northeast Siberia and Alaska; **b**, spatial scale plant richness to range-size relationship based on the modern time-slice (2000–0 years) from sedimentary ancient DNA; **c**, plant richness to range-size relationship based on modern plant taxa occurrence in 200 km x 200 km grid cells in northeast Siberia and Alaska; **d**, comparison of the range-sizes of the same taxa (present in both the 200 km x 200 km grid cells centred on the seven lakes and the 200 km x 200 km grid cells within the northeast Siberia and Alaska region) based on two methods: summing the grid cells centred on the seven lakes, and summing the grid cells within the northeast Siberia and Alaska region.

Reviewer comment: Fig. 4: If Figure 4 appears in the text before Figure 3, it would be advisable to switch their numbering.

Response: agree, revised.

Reviewer comment: Fig. 3: Diverging colours could enhance the clarity of range-size in Figure 3.

Response: the transparency has been adjusted to increase the plot clarity.

New figure: Line 684-686:

Figure 4: Plant taxa richness to range-size relationships (with 95% confidence intervals) per 5,000-year time window over the last 30,000 years inferred from lake sediment ancient DNA collected from northeast Siberia and Alaska. Coloured points show the richness and mean range-sizes of the 1,000-year time-slice samples (100 resampling iterations). The mean range-size is determined by calculating the average number of lakes occupied.

Reviewer comment: Fig. 4: Adding a Northern Hemisphere temperature reconstruction not inferred from vegetation to Figure 4g for comparison with pollen-based regional temperature would be beneficial.

Response: agree, revised.

New figure: Line 661-682:

Figure 3: Changes in the plant richness to range-size relationship and potential drivers of the relationship. a, plant richness change over time; **b**, range-size change over time; **c**, the plant richness to range-size relationship shifts from positive to negative at the transition

from the glacial to interglacial (the relationship was calculated based on the richness and range-size, colour gradient is similar to that in Figure 4); **d**, biotic environmental heterogeneity over time; **e**, cushion plant abundance (%) over time based on the mean value of 100 resampling results of our sedimentary ancient DNA dataset from seven lakes in northeast Siberia and Alaska; **f**, tree plant abundance (%) over time based on the mean value of 100 resampling results of our sedimentary ancient DNA dataset from seven lakes in northeast Siberia and Alaska; **g**, reconstructed mean annual temperature (°C, with 95% confidence intervals) based on ten pollen sites in the study region using the weighted averaging partial least squares (WAPLS) method, over the past 30,000 year. Some sites are the same as the sedaDNA data sites. **h**, temperature index represented by oxygen isotope values ($\delta^{18}\text{O}$) over the past 30,000 years from the North Greenland Ice Core Project (GRIP)⁷⁰, higher values correspond to higher temperatures. To compare with the richness to range-size relationship (calculated based on 5,000-year (5 ka) intervals for each time window), and to smooth the data to eliminate noise, all the data are shown in 5,000-year running time windows.

Reviewer #3 (Remarks to the Author):

In this research, the authors used sedimentary ancient DNA in cores across Siberia and Alaska to examine how the relationship between plant richness and range size changes through time. In the present day, this relationship is nearly ubiquitously negative (sites with higher species richness have smaller average range sizes); however, the authors found that this negative pattern has not persisted through time, linking changes in the richness-range size relationship to positive interactions/facilitation by cushion plants and negative interactions/exclusion by larger tree species. The results of this research (that the richness-range size relationship has not always been negative) are very interesting and could provide exciting new insight into how communities are structured across space and time. However, I have a few concerns I'd like to see addressed before I can fully agree with the mechanism/hypothesis put forward to explain this temporal variance.

First, I believe a moderate expansion on the methods is necessary to ensure replicability. The data collection and management are both very well documented, but, aside from the network analysis, the analyses conducted on the richness-range size relationship need a bit more information. In particular, it was unclear how the actual relationship was calculated (I assume a linear model of some sort, and correlations were mentioned in line 137, but I'm unsure how the uncertainty estimates were generated), and why the sliding window analysis (not mentioned until Lines 137) was necessary. Similarly, although the GBIF analyses is mentioned in the methods, it was unclear in the remainder of the paper (and especially in Figure 2b) what was done. Expanding and clarifying the methods would go a long way to make the research easier to understand and interpret.

Second, I very much appreciate the inclusion of the three hypotheses for the different relationships expected, and believe these are necessary for the interpretation of the paper. However, the justification behind the expected patterns based on the three hypotheses is often unclear. I think that my confusion might stem from the fact that the manuscript bounces back and forth between dealing with temporal trends (i.e., the time series patterns of the richness-range size relationship) and spatial trends (i.e., whether the range size-richness patterns are positive or negative at a single time point). I expand on the places I was unclear about in my line-level comments.

Response: We thank reviewer #3 for their constructive comments.

Abstract:

Reviewer comment: Line 15: Unclear what is meant by “inherent temporal dimensions”

Response: species accumulation time difference exists across spatial scales, the relationship in space is the complex of temporal and the spatial signals.

New text: Line 15-17: However, the complexity of the underlying mechanism limits its applicability for future environmental conservation or geographical range prediction.

Reviewer comment: Line 17: I would just say “richness-range size relationship”; “richness-mean range-size” sounds like you’re taking the mean of richnesses instead of the range sizes. Alternately, “the relationship between richness and mean range size” as before.

Response: We changed it to “richness to range-size relationship” in the whole text.

Reviewer comment: Line 18: the major factors of what? The identity of the richness-range size relationship or its change through time? This difference drives most of my confusion with regards to the interpretation of the hypotheses. See my comments on lines 53-56 too.

Response: the major factors of richness to range-size relationship, revised.

New text: Line 20-22: whether plant speciation, environmental heterogeneity, or plant interactions are the major factors of the relationship within the ice-shield-free northeast Siberia plus Alaska region.

Reviewer comment: Line 24: see comments on line 18

Response: done.

Introduction:

Reviewer comment: Line 35: either “nearly ubiquitous across space” or “nearly ubiquitously observed across space”

Response: agree, revised.

New text: Line 37-38: A negative relationship between richness and mean range-size is nearly ubiquitously observed across space.

Reviewer comment: Line 39: But mean range size itself doesn't expand, it's often the fact that smaller-ranged species are going extinct/reducing their ranges that leads to an increase in mean range size at a site.

Response: revised, this study avoids the potential extinction effect by using taxa (ASVs) that have a 100% match with the database, see the later explanation.

New text: Line 41-42: It is of particular interest whether the observed species richness decline in the context of global change ⁴ is associated with an increase in mean range-size.

Reviewer comment: Line 45: “10⁶ and 10⁵ years” is a perfectly fine way to write this, but maybe “thousands to millions of years” would be clearer? Definitely don't feel obligated to change it, though, it's fine as it is.

Response: revised.

Reviewer comment: Line 46-53: The example provided here (the latitudinal diversity gradient) occurs on a global scale: what about the same relationships on local or regional scales, where there are still differences in diversity from speciation/extinction rates? That would get at the next hypothesis (heterogeneity), but because these processes act on very different spatial scales in addition to temporal scales, I'm not sure they're directly comparable. Maybe the differences in scale are the point, though – is this research examining the differences in spatial and temporal scale of these drivers explicitly?

Response: Sorry for the misunderstanding, in the spatial scale (see Figure1), different drivers in different spatial scale drive the relationship (Figure1a, 1c, 1e). They all show negative relationship in different spatial scale.

Sorry for the misunderstanding, in the spatial scale (see Figure 1), there are different drivers of the relationship depending on the spatial scale (Figure 1a, 1c, 1e). All relationships, however, are negative at the spatial scale, regardless of the drivers.

This study only focuses on the temporal scale in the Siberia and Alaska region, to see if there are relationships between richness and range size change. If we apply the mechanism from the spatial scale to the temporal analyses, we assume no relationship for the speciation-extinction mechanism (due to the small influence of speciation and extinction (1b)). We assume a negative relationship for the heterogeneity mechanism (1d), and relationship change for the interaction (1f).

We revised the text to clarify and supply a calculation process (Figure 7) in the Methods part.

New text: Line 48-67: First, long-term (millions of years) temperature-related speciation and species extinction may result in negative richness to mean range-size relationships at the spatial scale^{7,8}. Higher temperatures increase metabolic rates and promote speciation rates⁹. Within a high speciation region, new species are initially constrained within a small range, resulting in regions with high species richness and smaller range-sizes^{8,10} as exemplified by the equatorial region¹¹. In contrast, low temperatures and recurrent extensive glaciation in high-latitude regions have resulted in less speciation and increased the risk of regional species extinctions¹². Thus, speciation and extinction progress along the temperature gradient results in a negative richness to range-size relationship in space (Figure 1a).

However, this hypothesis applies to the global pattern, and is not solely attributed to temperature differences across space but also to the accumulation of the difference over millions of years. Within a specific region and millennia scale, the impact of speciation and extinction may be limited.

Hence, **to shift the perspective to a temporal scale over millennia**, richness and range changes are not driven by speciation. Moreover, without considering extinction, there is no expected relationship between richness and range-size across time (Figure 1b).

New figure: Line 617-637:

Figure 1: Evaluating three hypotheses of how species richness relates to range size over space (previous study) and time (this study), using a hypothetical richness increase scenario. Coloured polygons represent the distribution range of different plant taxa. Under the speciation hypothesis, regions experiencing high speciation rates over millions of years generate taxa with a constrained range, resulting in a negative richness to range-size relationship (a), in the temporal scale, no speciation occurs on millennial time scales, neither during the glacial period nor the interglacial period. The richness increases are not driven by speciation, and range-size remains stable or changes without a discernible pattern, thus, there is no richness to range-size relationship (b). Under the environmental heterogeneity hypothesis, the temporal scale pattern (d) mirrors the spatial scale pattern (c). Spanning both glacial and interglacial periods, heterogeneous environments favour a high richness of taxa with narrow ecological niches and limited geographic ranges, leading to a stable negative richness to range-size relationship (d). Under the plant interaction hypothesis, widespread negative richness to range-size relationships are observed on a spatial scale, attributed to negative interactions (e). Unlike the spatial scale result, in the temporal scale, a positive interaction during the glacial period expands taxa range-sizes (f), resulting in a positive richness to range-size relationship; while a negative interaction during the interglacial period

constrains the range-sizes of the taxa, resulting in a negative relationship. The relationship shifts from positive during the glacial towards negative during the interglacial (f).

New figure: Line 705-719:

Figure 7: Schematic diagram of the richness to range-size relationship calculation. On the right, lakes are represented by a red dot, and remain the same for the past 30 timeslices. The study region was divided into 200 km x 200 km grid cells, which were used to evaluate the taxa ranges. On the left, A, B, C, and D represent different taxa that are shown in different coloured circles. Taking timeslice1 as an example, taxa distribution is available from the sedaDNA data (**step 1**). For the AOO (Area of Occupancy) method, the range of a specific taxon corresponds to the number of lakes in which the taxon occurred (**step 2**). For the EOO (Extent of Occurrence) method, the range was calculated as the sum of the grid cell areas overlapping with the convex hull spanning the lakes in which a specific taxon occurred (**step 2**). The richness in every timeslice is the total number of taxa types of all lakes, which is 4 in timeslice1; the mean range-size refers to the average range-size of all taxa (**step 3**). After computing the richness and mean range-size for each timeslice, a linear regression model was run to show the relationship between these variables within 5,000-year time-windows (**step4**).

Reviewer comment: Lines 53-56: These statements are unclear for a few reasons: First, the main points of the paper deal with changes back through time, but this statement talks about future dynamics and human-relevant timescales, which are not really examined in the paper. In addition, is there reason to believe that species extinctions (especially in a regional area like Siberia+Alaska) cannot happen on the timescales studied? Finally, the justification for the speciation hypothesis is entirely spatial (i.e., areas of higher vs. lower species richness), so I'm confused where the interpretations for temporal changes in this relationship come from (see comments on line 18).

Also, if speciation rates happen faster when it is warmer and extinction rates are slower (as mentioned in lines 46-47), causing a more strongly negative richness-range size relationship, wouldn't you expect negative relationships in the Holocene and less negative relationships in the glacial period (exactly what is found in the results)? I just think a little bit more explanation would really help the interpretation of this hypothesis.

Response: We have removed future dynamics and human-relevant timescales.

We apologise for the incorrect statement suggesting that species extinctions cannot occur within the studied timescales. Our analysis was conducted using a database constructed from modern plants, and only ASVs with 100% correspondence to the database were included, which was believed to preclude extinction effects.

Additionally, we test that ASVs with 100% correspondence to the database exist in both the Glacial and the Holocene period, which was assumed to remove the extinction effect. The results show that the relationship did not change (see below **Supplementary Figure 7**)

To justify the relationship, we have added the above-mentioned Figure 1 and Figure 7 to show the process. If there are still any unclear parts, we appreciate you pointing them out, and we will provide further clarification.

Regarding speciation, our hypothesis assumes that a long-term scale is also necessary. We have revised the text to incorporate this time information.

New figure in Supplementary: Line 245-251: Supplementary Figure 7: (using the ASVs that were present both during the Glacial period and the Holocene period)

Supplementary Figure 7: Plant taxa richness to range-size relationships (with confidence intervals) per 5,000-year time windows over the last 30,000 years, inferred from the taxa that occur in both the glacial and Holocene periods across northeast Siberia and Alaska. Coloured points show the richness and mean range sizes of the 1,000-year time-slice samples (100 resampling iterations). The mean range size is determined by calculating the average number of lakes occupied.

New text: Line 48-67: First, long-term (millions of years) temperature-related speciation and species extinction may result in negative richness to mean range-size relationships at the spatial scale ^{7,8}. Higher temperatures increase metabolic rates and promote speciation rates ⁹. Within a high speciation region, new species are initially constrained within a small range, resulting in regions with high species richness and smaller range-sizes ^{8,10} as exemplified by the equatorial region ¹¹. In contrast, low temperatures and recurrent extensive glaciation in high-latitude regions have resulted in less speciation and increased the risk of regional species extinctions ¹² Thus, speciation and extinction progress along the temperature gradient results in a negative richness to range-size relationship in space (Figure 1a).

However, this hypothesis applies to the global pattern, and is not solely attributed to temperature differences across space but also to the accumulation of the difference over millions of years. Within a specific region and millennia scale, the impact of speciation and extinction may be limited.

Hence, **to shift the perspective to a temporal scale over millennia**, richness and range changes are not driven by speciation. Moreover, without considering extinction,

there is no expected relationship between richness and range-size across time (Figure 1b).

New text: Line 113-115: If million years of speciation and species loss are the primary factors, when applying this pattern to millennia studies, especially after eliminating the potential impact of extinction, no richness to range-size relationship would be expected.

New text: Line 181-184: Given that we detect significant richness to mean range-size relationships for almost all time-windows in our study over the last 30,000 year ka, and that the potential extinction effect was eliminated by using taxa with 100% assignment to the database, we conclude that processes operating on millennial time-scales also influence the relationship.

Reviewer comment: Line 57: Very well-reasoned and good justification! As mentioned before, environmental heterogeneity is a smaller scale (spatial and temporal) than the previous hypothesis, but maybe that's the point.

Response: agree and explain in the "Line 46-53 comments".

Reviewer comment: Line 68: I might rearrange this sentence to ensure that the stress-gradient hypothesis is defined before talking about the hypothesis on how it could be applied to the case study of this manuscript. Here's one possible suggestion: "The stress-gradient hypothesis (cite) suggests that in less stressful environments, high richness can _____. In contrast, _____", and then bringing up how this hypothesis might be applied to temporal change.

Response: agree and revised.

New text: Line 80-88: The stress-gradient hypothesis ^{19,20} suggests that in less stressful environments, negative interactions are promoted, while in stressful environments, positive interactions should be more common. In line with the stress-gradient hypothesis, it may be hypothesised that in less stressful environments, high richness can lead to increased competition and predation (negative interactions), limiting growth rate ²¹ and reducing range-sizes ³, resulting in a negative richness to range-size relationship. In contrast, environmental stress can promote positive interactions among taxa allowing for mutually wide ranges ^{19,20,22,23}, which would be reflected by a positive richness to range-size relationship.

Reviewer comment: Line 75-77: This hypothesis assumes that the species are less stressed during times of warming, which may well be the case, but some very cold-adapted species (especially in the lower latitudes) might be more stressed in warmer temperatures than cold ones.

Response: agree, and revised to “from the stressful tundra environment to the relatively less stressful taiga environment”.

New text: Line 90-94: In the temporal domain, an environmental stress release—such as from the stressful tundra environment during the glacial period to the relatively less stressful taiga environment during the Holocene—may lead to a shift from a positive to a negative richness to range-size relationship (Figure 1f).

Reviewer comment: Line 80-86: This is very clear and understandable!

Response: Thanks.

Reviewer comment: Line 88-91: You may want to split this sentence into two (maybe at the “,which,”)

Response: agree, revised.

New text: Line 105-109: Furthermore, previous studies from the area have yielded high-quality sedimentary ancient DNA (sedaDNA) records. Compared to traditional pollen analysis, sedaDNA can identify past plants to a higher taxonomic resolution, which allows the reconstruction of past plant richness and distribution changes at a reasonable taxonomic level ²⁸.

Reviewer comment: Line 92: change “exploit” to “use”

Response: done.

Reviewer comment: Line 95-98: See my comments on lines 18, 53-56: I think it is possible to see a significant relationship that changes through time just based on small-ranged species going extinct.

Response: revised.

New text: Line 113-115: If million years of speciation and species loss are the primary factors, when applying this pattern to millennia studies, especially after eliminating the potential impact of extinction, no richness to range-size relationship would be expected.

Reviewer comment: Line 99: “consistently” instead of “stably”

Response: done.

Results and Discussion:

Reviewer comment: Lines 108-118: Very well explained!

Response: Thanks.

Reviewer comment: Line 122: Although Figure 2 alludes to it, I believe this is the first mention of the GBIF analyses. Moving the subsequent sentence (starting with “Furthermore, the range size of GBIF-derived plants”) before this sentence would help a lot in clarifying what the GBIF analyses are. Maybe even the sentences in the Methods about this (Lines 281-285) would be helpful here.

Response: agree, revised.

New text: Line 150-160: Modern plant information for the northeast Siberia and Alaska region was derived from GBIF, and range-size was determined based on the sum of occupied 200 km x 200 km grid cells. In space, plant richness and range-size among grid cells show negative relationships ($r = -0.37$, $p < 0.01$) (Figure 2c), as observed in the DNA results (Figure 2b).

Furthermore, for the same GBIF-derived taxa, their range-sizes were calculated using AOO method, based on grid cells centred around the seven lakes, and the larger Siberian and Alaskan regions. The results based on two regions reveal a high correspondence pattern (Figure 2d), validating the representation of plant taxa range-sizes based on the seven lakes.

Reviewer comment: Line 126-129: Cool! And good to know that the method of calculation doesn't appear to affect your results!

Response: Thanks.

Reviewer comment: Line 137: Why do 5-ka time windows for the sliding-window analysis? How does the breadth of this window affect your analyses (e.g., 5ka vs 2ka vs 10ka windows)?

Response: revised.

New text: Line 680-682: To compare with the richness to range-size relationship (calculated based on 5,000-year (5 ka) intervals for each time window), and to smooth the data to eliminate noise, all the data are shown in 5,000-year running time windows.

Supplementary file Line 195-199: We tested plant taxa richness to range-size relationships across various time windows, ranging from 10,000-year (10ka) to 2,000-years (2ka), revealing a consistent pattern (Supplementary Figure 4). To incorporate

more time slices, thereby reducing noise from limited data, and to examine the relationship for the glacial and Holocene periods separately, 5,000-year (5ka) time window is used in the main text.

Supplementary Figure 4 Line 221-227:

Supplementary Figure 4: Plant taxa richness to range-size relationships for time windows ranging from 10,000-year (10 ka) to 2,000-year (2 ka), over the last 30,000 years, inferred from lake sedimentary ancient DNA collected from northeast Siberia and Alaska region. Coloured points show the richness and mean range sizes of the 1,000-year time-slice samples (100 resampling iterations). The mean range size is determined by calculating the average number of lakes occupied.

Reviewer comment: Line 137: revise to “for the glacial period (30-11 ka) and negative...”

Response: done.

New text: Line 171-173: Correlation analyses find positive temporal relationships between plant richness and range-size within 5,000-year time-windows for the glacial period (30–11 ka) and negative relationships for the Holocene (11–0 ka, Figure 3c, Figure 4).

Reviewer comment: Line 138: Were the p-values in Supplementary Table 2 (and shown in Figure 3) corrected for multiple comparisons? If not, I’d suggest using the `p.adjust()` function in `r` to do so.

Response: done.

New text Line 255-260: Supplementary Table 2 and Supplementary Table 3.

Reviewer comment: Lines 146-148: See comments on line 18, Lines 53-56

Response: revised.

New text: Line 181-187: Given that we detect significant richness to range-size relationships for almost all time-windows in our study over the last 30,000 year, and that the potential extinction effect was eliminated by using taxa with 100% assignment to the database, we conclude that processes operating on millennial time-scales also influence the relationship. We reject hypothesis (1), which posited that temperature-related speciation and glaciation-related species extinctions are the sole major factor shaping the richness to range-size relationship.

Reviewer comment: Lines 175-176: What does a positive vs. negative relationship mean in this case? Effect size > 0 vs. effect size < 0 ? I think you’d want to define a threshold (say, $R > 0.2$ or < -0.2), because a slope of, say 0.000001 is not particularly biologically significant.

Response: We previously used “ $R > 0$ ” and “ $R < 0$ ” to represent the positive and negative relationships. We revised it to “ $R > 0.2$ ” and “ $R < -0.2$ ”.

New text: Line 219-224: To test the plant interaction hypothesis, two generalised linear models were constructed, with richness to range-size relationship as binomial response variables (positive for the correlation $R > 0.2$, and negative for the correlation $R < -0.2$) and cushion plant abundance and tree abundance as the explanatory variables (Figure 5).

Reviewer comment: Line 177: This is the first time that cushion plants are mentioned – I think parts of the excellent explanation provided in Lines 188-208 should be moved before this paragraph to prime the reader for the ecological role of cushion plants and why they are important for the study.

Response: agree, we have changed the sentence order.

New text: Line 218-219: Serving as nurse plants, cushion plants inhabiting arctic or alpine regions exhibit a low-growing, mat-forming growth pattern, which supports a positive interaction with other plants⁴⁰.

Reviewer comment: Line 182-185: Very interesting, I love the idea of using positive network interactions! I know this is mentioned in the methods, but maybe make it clear that the internal edges are all positive interactions. Also, I'd be interested to know about the proportion of positive interactions, in addition to the absolute number of them.

Response: we add the proportion of positive interactions (positive interaction/positive interaction+negative interaction).

New text: Line 228-234: The network, constructed from **positive** pairwise plant taxa correlations ($r > 0.6$), delineates two communities (Figure 6). The glacial community, incorporating cushion plant taxa, comprises 11 nodes and 30 **positive** internal edges; a **68.78% positive link proportion**. In contrast, the Holocene community, which includes tree taxa, also consists of 11 nodes, but 17 **positive** internal edges; a **38.64% positive link proportion**. For the glacial community, more links are detected compared with the Holocene community, which indicates more associations among plant taxa and thus a higher possibility of positive plant interactions.

Reviewer comment: Lines 216-219: Revise to something more like “However, this study relies on the correlation of selected drivers. To uncover causal evidence, ...” Also, “the incorporation of a model” is somewhat vague, I think it could be improved by adding one or two specific examples of what can be done to uncover causal evidence.

Response: agree, revised.

New text Line 264-268: This study, however, is simplified by relying on the correlation of selected drivers to uncover causal evidence, and the incorporation of a model is

essential, such as an **individual-based model**⁵¹, a **multispecies model**⁵², or a **species distribution model**⁵³. Nevertheless, the implications of the shift in the richness to range-size relationship cannot be overemphasised.

Reviewer comment: Line 228: Maybe revise from “In analogy to the past” to “In the future” or something like that, where the future implications are highlighted as opposed to the past (which is what the main body of the paper discusses).

Response: agree, revised.

New text Line 277-288: In the future, to conserve tundra biodiversity effectively, attention should be towards the far northern tundra region. Within the context of climate change, colonisation in the present-day subarctic regions by trees and shrubs promotes the establishment of negative interactions, thereby constraining the range-size of the tundra taxa and hampering protection of the taxa. In the boreal region, certain understorey herb species with limited competitiveness will exhibit restricted range-sizes. Likewise, their ability to migrate northward with climate warming may be hindered due to their poorer competitive ability. In contrast, in far northern tundra areas dominated by harsh environments, positive interactions may expand the taxa ranges and enhance the protection of tundra taxa, for example, translocations of the endangered species. Additionally, careful measures should be taken to prevent the introduction of alien species, especially those mediated by human activity, as the positive interaction characteristic of these regions makes them more susceptible to alien taxa invasion.

Methods:

Reviewer comment: Lines 273-280: why 200 by 200 km grid cells? Does using different spatial resolutions change the results?

Response: revised.

New text: Supplementary file Line 200-205: During the calculation of the modern plant richness to range-size relationship across spatial scales, we tested this relationship (Supplementary Figure 5a, c, e, g) and compared taxon range sizes (Supplementary Figure 5b, d, f, h) using four grid cell sizes, ranging from 100 km × 100 km to 500 km × 500 km. The results consistently exhibited the same pattern. To mitigate the influence of any limited data resolution in GBIF, we use 200 km × 200 km grid cells in the main text.

Supplementary Figure 5 Line 228-238:

Supplementary Figure 5: The modern plant richness to range-size relationship across space in the northeast Siberia and Alaska region, based on modern plant taxa occurrences

in grid cells of four sizes from 100 km × 100 km grid cells to 500 km × 500 km grid cells (a, c, e, g). For each taxon, the range-size is defined as the sum of grid cells in which it occurred, for one grid cell, richness was determined by the number of taxa types, and the mean range-size was the average of all taxa ranges. Comparison of the same taxa range-size between two calculation methods: based on the sum of occupied grid cells centred on the seven lakes, and the sum of occupied grid cells in northeast Siberia and Alaska region, for four grid cell sizes ranging from 100 km × 100 km to 500 km × 500 km (b, d, f, h).

Reviewer comment: Lines 286-293: Needs more citations (e.g., Baselga et al. 2010 GEB for the sentence ending on line 291). Also, cite the r package used.

Response: done.

Reviewer comment: Lines 287-288: some methods of beta diversity are calculated like this, but the way you're calculating beta diversity (with Jaccard turnover) is not.

Response: agree, and revised.

New text: Line 375-378: Beta diversity represents the dissimilarities of multiple sites, which constitutes two components: richness differences (or nestedness indicating the richness difference between sites) and species replacement or turnover^{66,67}.

Reviewer comment: Line 291: Very cool, I like the fact that you're accounting for the richness differences by using turnover specifically instead of both turnover and nestedness.

Response: thanks.

Reviewer comment: Lines 294-298: Would it be possible to compare the proportion of cushion/tree taxa to the number/proportion of positive interactions directly? It appears from Figure 6 and from the results that they constitute two different communities, one with many more positive interactions than the other, but I would like to see that explicitly stated in the paper.

Also, to clarify: The relative abundance was calculated by the number of taxa (i.e., the diversity) of cushion plants/trees, or the number of individual observations? If it's the number of taxa, I'm not sure that gets at dominance, just diversity.

Response: revised.

New text: Line 229-234: The glacial community, incorporating cushion plant taxa, comprises 11 nodes and **30 positive** internal edges; a **68.78% positive link proportion**. In contrast, the Holocene community, which includes tree taxa, also consists of 11 nodes, but **17 positive** internal edges; a **38.64% positive link**

proportion. For the glacial community, more links are detected compared with the Holocene community, which indicates more associations among plant taxa and thus a higher possibility of positive plant interactions.

Line 387-389: The relative abundance was calculated as the ratio between the number of cushion-plant/tree taxa observations and the sum of all taxa observations within the time slice.

Reviewer comment: Line 299-303: Using the pollen assemblages to reconstruct the temperature is going to be circular, given that the study examines changes in plant assemblages through time. This isn't to say you should remove it; instead, I'd suggest comparing both the pollen-reconstructed temperature and temperature derived from a different method (e.g., global circulation models) and using both in the manuscript and in Figure 4.

Response: agree, revised.

New figure: Line 661-682:

Figure 3: Changes in the plant richness to range-size relationship and potential drivers of the relationship. **a**, plant richness change over time; **b**, range-size change over time; **c**, the plant richness to range-size relationship shifts from positive to negative at the transition

from the glacial to interglacial (the relationship was calculated based on the richness and range-size, colour gradient is similar to that in Figure 4); **d**, biotic environmental heterogeneity over time; **e**, cushion plant abundance (%) over time based on the mean value of 100 resampling results of our sedimentary ancient DNA dataset from seven lakes in northeast Siberia and Alaska; **f**, tree plant abundance (%) over time based on the mean value of 100 resampling results of our sedimentary ancient DNA dataset from seven lakes in northeast Siberia and Alaska; **g**, reconstructed mean annual temperature (°C, with 95% confidence intervals) based on ten pollen sites in the study region using the weighted averaging partial least squares (WAPLS) method, over the past 30,000 year. Some sites are the same as the sedaDNA data sites. **h**, temperature index represented by oxygen isotope values ($\delta^{18}\text{O}$) over the past 30,000 years from the North Greenland Ice Core Project (GRIP)⁷⁰, higher values correspond to higher temperatures. To compare with the richness to range-size relationship (calculated based on 5,000-year (5 ka) intervals for each time window), and to smooth the data to eliminate noise, all the data are shown in 5,000-year running time windows.

Reviewer comment: Line 324: List the packages used (igraph, betapart, others) with citations either here or in a supplement.

Response: done.

New text: Supplementary text Line 206-209:

All data analyses in the main text and supplementary texts were done in R version 4.3.2¹⁰, with packages vegan¹¹, rgbif^{12,13}, stats¹⁰, arm¹⁴, betapart¹⁵, rioja¹⁶, Hmisc¹⁷, igraph package^{18,19}, dplyr²⁰, data.table²¹, tidyr²², sf^{23,24}, ggplot2²⁵, viridis²⁶, cowplot²⁷, ggpubr²⁸, crayon²⁹, palaeoSig³⁰, and tidyverse³¹.

Reviewer comment: Figure 1: I appreciate the conceptual figure of the different hypotheses and how they might look along the landscape. However, I think this figure falls into the same trap of looking at temporal patterns vs. spatial patterns: For example, if I am reading the hypotheses correctly from the introduction, the speciation hypothesis would cause spatial patterns in richness/range size, but not temporal patterns. Figure 1c appears to show that there would be no relationship whatsoever if the speciation hypothesis holds. The other two hypotheses are very well-described in the figure, though! Maybe adding a separate column of figures like 1c, 1e, 1g, and 1i looking at the hypothesized relationships in the interglacial period and change the others to the hypothesized relationships in the glacial period specifically would help to clarify.

Response: we revised the conceptual figure.

New Figure Line 617-638:

Figure 1: Evaluating three hypotheses of how species richness relates to range size over space (previous study) and time (this study), using a hypothetical richness increase scenario. Coloured polygons represent the distribution range of different plant taxa. Under the speciation hypothesis, regions experiencing high speciation rates over millions of years generate taxa with a constrained range, resulting in a negative richness to range-size relationship (a), in the temporal scale, no speciation occurs on millennial time scales, neither during the glacial period nor the interglacial period. The richness increases are not driven by speciation, and range-size remains stable or changes without a discernible pattern, thus, there is no richness to range-size relationship (b). Under the environmental heterogeneity hypothesis, the temporal scale pattern (d) mirrors the spatial scale pattern (c). Spanning both glacial and interglacial periods, heterogeneous environments favour a high richness of taxa with narrow ecological niches and limited geographic ranges, leading to a stable negative richness to range-size relationship (d). Under the plant interaction hypothesis, widespread negative richness to range-size relationships are observed on a spatial scale, attributed to negative interactions (e). Unlike the spatial scale result, in the temporal scale, a positive interaction during the glacial period expands taxa range-sizes (f), resulting in a positive

richness to range-size relationship; while a negative interaction during the interglacial period constrains the range-sizes of the taxa, resulting in a negative relationship. The relationship shifts from positive during the glacial towards negative during the interglacial (f).

Figure 2:

Figure 2a: I don't think the coordinates along the edge of map box are necessary with the projection: because it's close to polar, the latitude and longitude lines are not perpendicular to the edges of the map box or to each other. You could add longitude lines in the map if you'd like, but I don't think the map needs it.

Also, (and this applies to Figure 2c as well), I would color-code the different sites so it's clear where each site is and the richness and mean range size of each. I only see 6 out of the 7 sites -- I assume Raachuagytygyn and Iirney are very close together, so I would jitter the map a little bit so that the reader can clearly see both.

Figure 2b: This is modern, correct? I would make it very clear that it is modern range sizes.

Figure 2c: Color code the different sites. This will also help to distinguish this plot from Figure 2d, which is (I believe) modern occurrence data.

Figure 2d: This is modern, correct? I would make it very clear that it is modern range sizes.

Caption: instead of saying "(by counting the number of grid cells in which each taxon occurs)", you can say "...range size of selected taxa based on the sum of occupied 200x200 km grid cells"

Response: revised.

New figure: Line 640-653:

Figure 2: Plant richness to range-size relationship across space. **a**, Location of the seven lakes in northeast Siberia and Alaska; **b**, spatial scale plant richness to range-size relationship based on the modern time-slice (2000–0 years) from sedimentary ancient DNA; **c**, plant richness to range-size relationship based on modern plant taxa occurrence in 200 km × 200 km grid cells in northeast Siberia and Alaska; **d**, comparison of the range-sizes of the same taxa (present in both the 200 km x 200 km grid cells centred on the seven lakes and the 200 km x 200 km grid cells within the northeast Siberia and Alaska region) based on two methods: summing the grid cells centred on the seven lakes, and summing the grid cells within the northeast Siberia and Alaska region.

Figure 3:

This is a beautiful figure!

Caption:

Line 528: I would add how the average range size is calculated (mean number of lakes occupied, right?)

Line 528: Typo “coloured”

Response: agree, revised.

New figure: Line 683-689 :

Figure 4: Plant taxa richness to range-size relationships (with 95% confidence intervals) per 5,000-year time window over the last 30,000 years inferred from lake sediment ancient DNA collected from northeast Siberia and Alaska. Coloured points show the richness and mean range-sizes of the 1,000-year time-slice samples (100 resampling iterations). The mean range-size is determined by calculating the average number of lakes occupied.

Figure 4: Good! The subheadings of the figures might be better outside the boxes, because at the moment they are a little hard to see.

Response: agree, revised.

New figure: Line 661-682 :

Figure 3: Changes in the plant richness to range-size relationship and potential drivers of the relationship. a, plant richness change over time; **b**, range-size change over time; **c**, the plant richness to range-size relationship shifts from positive to negative at the transition

from the glacial to interglacial (the relationship was calculated based on the richness and range-size, colour gradient is similar to that in Figure 4); **d**, biotic environmental heterogeneity over time; **e**, cushion plant abundance (%) over time based on the mean value of 100 resampling results of our sedimentary ancient DNA dataset from seven lakes in northeast Siberia and Alaska; **f**, tree plant abundance (%) over time based on the mean value of 100 resampling results of our sedimentary ancient DNA dataset from seven lakes in northeast Siberia and Alaska; **g**, reconstructed mean annual temperature ($^{\circ}\text{C}$, with 95% confidence intervals) based on ten pollen sites in the study region using the weighted averaging partial least squares (WAPLS) method, over the past 30,000 year. Some sites are the same as the sedaDNA data sites. **h**, temperature index represented by oxygen isotope values ($\delta^{18}\text{O}$) over the past 30,000 years from the North Greenland Ice Core Project (GRIP)⁷⁰, higher values correspond to higher temperatures. To compare with the richness to range-size relationship (calculated based on 5,000-year (5 ka) intervals for each time window), and to smooth the data to eliminate noise, all the data are shown in 5,000-year running time windows.

Figure 5: This is a very clear figure and it really shows the importance of cushion plants/tree proportion in driving this relationship. See my comments on Line 175-176.

Response: done.

New Figure: Line 692-696:

Figure 5: Binomial regression plot ($p < 0.001$) of the richness to mean range-size relationship ($R > 0.2$ for positive relationship, $R < -0.2$ for negative relationship) with relative abundance of cushion plant taxa (%) and tree plant taxa (%), based on the median value of a 5,000-year running time-window for every resampling round.

Figure 6:

Good! Maybe use different colors for the two communities, because they were previously used in Figures 3,4 for cushion plants vs. trees. Also, maybe distinguish the tree and cushion plant taxa on the graphs themselves (e.g., putting a black border around the cushion plant nodes). Finally, I think the overall graphs (including both positive and negative associations) should be in the supplements.

Response: thank you, revised.

New Figure: Line 698-704:

Figure 6: Positive correlation network of plant groups. Plant families (cushion-plant taxa are outlined in black, including Boraginaceae, Caryophyllaceae, Saxifragaceae, and Brassicaceae; tree taxa are outlined in purple, including Betulaceae and Pinaceae) are represented by nodes, where the size of the node represents the number of links (node degree). Edges indicate positive correlations between plant groups. The Holocene and glacial communities are shown separately.

Supplementary figure : Line 239-244:

Supplementary Figure 6: Correlation network of plant groups. Plant families are represented by coloured nodes, where the size of the node represents the number of links (node degree), red nodes represent taxa belonging to the glacial community, and blue nodes represent the taxa belonging to the Holocene community. Blue edges indicate negative correlations between taxa, and red edges indicate positive correlations between taxa.

REVIEWERS' COMMENTS

Reviewer #1 (Remarks to the Author):

Thank you very much for the clear rebuttals, and I am happy with the corrections made. I have no further comments.

Reviewer #3 (Remarks to the Author):

The authors have significantly modified the manuscript according to my previous comments, and I found that the clarity of the manuscript was significantly improved. In particular, the highlighting of the conceptual differences between examining the range-size to richness relationship across space vs. across time really helped my understanding of the project. I have a few additional comments that I think could further clarify and strengthen the manuscript. To note, all of my comments below reference lines from the Track Changes document provided.

General Comment 1:

I appreciate the distinguishing between examining the relationship in space vs. through time, and the new figure really helps out in clarifying the pattern, but I think the use of "spatial scale" and "temporal scale" in this context is a little confusing. When I think of spatial /temporal scale, I think of the spatial/temporal extent or resolution of the study, not the axis through which the relationship is examined. This conflation actually occurs in the manuscript (Line 63-64): "within a specific region and millennia scale...to shift the perspective to a temporal scale". A few ideas for making this concept clearer might be to rename to "domain" (to modify Line 50, "...on range-size relationships in the spatial domain", e.g., Line 75), or, for a simpler construction: "across space" vs. "through time" (e.g., "...on range size relationships across space", e.g., Line 67).

General Comment 2:

The additions to the supplements are excellent and provide much-needed context to the results – make sure each of the supplements are referenced in the manuscript proper (for example, you could add something like "See Supplementary Figure 5 for an analysis of the grid cell size" after Lines 355-356), so that the work you've done and the results of the supplementary information are clear and easily accessed.

Line 52: Revise to "within a region with high speciation rates" or something similar.

Line 57: Maybe revise to "Variation in speciation and extinction rates along the temperature gradient"

Line 63: Maybe "the accumulation and extinction of species" instead of "the accumulation of the difference"? I'm not sure temperature differences themselves accumulated over millions of years, but rather species responses to them.

Line 89: Revise "is" back to "are"

Line 91: Good example! Is there a source/citation for tundra environments being more stressful than taiga environments across the species studied? It may be obvious, but it's possible (maybe even likely) that some cold and dry-adapted tundra plants would be more stressed in the taiga than in the tundra. If it's simply more stressful on aggregate (across the various species studied), that's perfect, but I still think a citation may be necessary to support it.

Line 101: add a comma after "regions"

Line 115: maybe emphasize the temporal aspect by saying something like "an insignificant richness to range-size relationship would be expected through time"

Lines 139-141: Maybe revise to "The mean range size ("range-size" for simplicity), refers to the average..." Also, I would move this sentence with the three sentences after it (Lines 141-145), so that first you discuss how range size was calculated and then you discuss averaging it.

Line 157: Revise to "using the AOO method"

Line 164-166: Provide the time periods for each of LGM, Early Late Glacial, and Early and late Holocene, similar to what you do in Line 169 for the range-size time series.

Line 178: See General Comment 1

Line 183: Just to make sure: the "SibAla_2023" database?

Line 184: Maybe revise to something like "influence the temporal relationship between richness and mean range size"?

Line 194: See Line 183

Line 195: Maybe something like this? "Therefore, speciation and extinction are not the primary drivers of the temporal richness to range size relationship on millennial time scales". You should emphasize that, although speciation and extinction are likely causes of the spatial relationship, they don't seem to matter for the temporal relationship.

Line 204: Revise "have" to "had", for the past? Unless the taxa you're talking about still survive and have a widespread distribution.

Line 204: "While" to "Although".

Line 208: Revise to "The highest values of heterogeneity were during the early..."

Line 219: Good! Maybe add a sentence before the one starting with "To test" that restates the hypothesis (i.e., that a high proportion of cushion plants leads to a more positive richness-range size relationship through time).

Line 260: I think this sentence (starting, "Overall, our study...") needs a little bit more of a link with the previous sentences: The start of the paragraph is all about the Stress gradient theory whereas this sentence looks at the plant interaction hypothesis put forward in this manuscript itself.

Line 307: Was the Bayesian age-depth model created for this project specifically or was an already existing model applied to these samples? If the latter, cite the model.

Figure 1: This figure is so much clearer now, well done! Just to make sure I understand it, this is an example where you'd have 11 timeslices in your sliding window, right? With 11 points in the plots on the right.

Line 619: Change "previous study" to "previous studies". Also, if there's a citation for it, maybe add it in here?

Line 623, 632, Figure Labels: See General Comment 1.

Response Letter

All changes were marked using the “**Track Changes**” function in the revised manuscript.

Reviewer #1 (Remarks to the Author):

Thank you very much for the clear rebuttals, and I am happy with the corrections made. I have no further comments.

Response: We are thankful to Reviewer #1 for their constructive comments.

Reviewer #3 (Remarks to the Author):

The authors have significantly modified the manuscript according to my previous comments, and I found that the clarity of the manuscript was significantly improved. In particular, the highlighting of the conceptual differences between examining the range-size to richness relationship across space vs. across time really helped my understanding of the project. I have a few additional comments that I think could further clarify and strengthen the manuscript. To note, all of my comments below reference lines from the Track Changes document provided.

Response: We are thankful to Reviewer #3 for their constructive comments. All responses below refer to **lines** from the new **Track Changes document**.

General Comment 1:

I appreciate the distinguishing between examining the relationship in space vs. through time, and the new figure really helps out in clarifying the pattern, but I think the use of “spatial scale” and “temporal scale” in this context is a little confusing. When I think of spatial /temporal scale, I think of the spatial/temporal extent or resolution of the study, not the axis through which the relationship is examined. This conflation actually occurs in the manuscript (Line 63-64): “within a specific region and millennia scale...to shift the perspective to a temporal scale”. A few ideas for making this concept clearer might be to rename to “domain” (to modify Line 50, “...on range-size relationships in the spatial domain”, e.g., Line 75), or, for a simpler construction: “across space” vs. “through time” (e.g., “...on range size relationships across space”, e.g., Line 67).

Response: agree, revised. **New text line 51, 63, 613-631.**

General Comment 2:

The additions to the supplements are excellent and provide much-needed context to the results – make sure each of the supplements are referenced in the manuscript proper (for example, you could add something like “See Supplementary Figure 5 for an analysis of the grid cell size” after Lines 355-356), so that the work you’ve done and the results of the supplementary information are clear and easily accessed.

Response: agree, revised. **New text line 142-144, 187, 339, 353.**

Reviewer Comment: Line 52: Revise to “within a region with high speciation rates” or something similar.

Response: done. **New text line 52-53.**

Reviewer Comment: Line 57: Maybe revise to “Variation in speciation and extinction rates along the temperature gradient”

Response: done. **New text line 57.**

Reviewer Comment: Line 63: Maybe “the accumulation and extinction of species” instead of “the accumulation of the difference”? I’m not sure temperature differences themselves accumulated over millions of years, but rather species responses to them.

Response: agree, revised. **New text line 60-61.**

Reviewer Comment: Line 89: Revise “is” back to “are”

Response: done. **New text line 86.**

Reviewer Comment: Line 91: Good example! Is there a source/citation for tundra environments being more stressful than taiga environments across the species studied? It may be obvious, but it’s possible (maybe even likely) that some cold and dry-adapted tundra plants would be more stressed in the taiga than in the tundra. If it’s simply more stressful on aggregate (across the various species studied), that’s perfect, but I still think a citation may be necessary to support it.

Response: We included three references, which demonstrate the temperature increased from the Glacial period to the Holocene. And the taiga environment is less stressful for most plants, which is shown by the plant richness and productivity increase. **New text line 87-90.**

Reviewer Comment: Line 101: add a comma after “regions”.

Response: done. **New text line 100.**

Reviewer Comment: Line 115: maybe emphasize the temporal aspect by saying something like “an insignificant richness to range-size relationship would be expected through time”.

Response: done. **New text line 113-114.**

Reviewer Comment: Lines 139-141: Maybe revise to “The mean range size (“range-size” for simplicity), refers to the average...” Also, I would move this sentence with the three sentences after it (Lines 141-145), so that first you discuss how range size was calculated and then you discuss averaging it.

Response: done. **New text line 137-141.**

Reviewer Comment: Line 157: Revise to “using the AOO method”.

Response: done. **New text line 152.**

Reviewer Comment: Line 164-166: Provide the time periods for each of LGM, Early Late Glacial, and Early and late Holocene, similar to what you do in Line 169 for the range-size time series.

Response: done. **New text line 158-164.**

Reviewer Comment: Line 178: See General Comment 1.

Response: revised. **New text line 171.**

Reviewer Comment: Line 183: Just to make sure: the “SibAla_2023” database?

Response: yes, revised. **New text line 175.**

Reviewer Comment: Line 184: Maybe revise to something like “influence the temporal relationship between richness and mean range size”?

Response: done. **New text line 176-177.**

Reviewer Comment: Line 194: See Line 183

Response: done. **New text line 186.**

Reviewer Comment: Line 195: Maybe something like this? “Therefore, speciation and extinction are not the primary drivers of the temporal richness to range size relationship on millennial time scales”. You should emphasize that, although speciation and extinction are likely causes of the spatial relationship, they don’t seem to matter for the temporal relationship.

Response: agree, revised. **New text line 187-188.**

Reviewer Comment: Line 204: Revise “have’ to “had”, for the past? Unless the taxa you’re talking about still survive and have a widespread distribution.

Response: agree, revised. **New text line 195.**

Reviewer Comment: Line 204: “While” to “Although”.

Response: done. **New text line 195.**

Reviewer Comment: Line 208: Revise to “The highest values of heterogeneity were during the early...”

Response: done. **New text line 197-198.**

Reviewer Comment: Line 219: Good! Maybe add a sentence before the one starting with “To test” that restates the hypothesis (i.e., that a high proportion of cushion plants leads to a more positive richness-range size relationship through time).

Response: done. **New text line 209-210.**

Reviewer Comment: Line 260: I think this sentence (starting, “Overall, our study...”) needs a little bit more of a link with the previous sentences: The start of the paragraph is all about the Stress gradient theory whereas this sentence looks at the plant interaction hypothesis put forward in this manuscript itself.

Response: agree, revised. **New text line 249-250.**

Reviewer Comment: Line 307: Was the Bayesian age-depth model created for this project specifically or was an already existing model applied to these samples? If the latter, cite the model.

Response: We cited the pre-existing Bayesian age-depth model. **New text line 289.**

Reviewer Comment: Figure 1: This figure is so much clearer now, well done! Just to make sure I understand it, this is an example where you’d have 11 timeslices in your sliding window, right? With 11 points in the plots on the right.

Response: Yes, each point represents a timeslice, and we just used 11 points randomly as an example.

Reviewer Comment: Line 619: Change “previous study” to “previous studies”. Also, if there’s a citation for it, maybe add it in here?

Response: done. **New text line 615.**

Reviewer Comment: Line 623, 632, Figure Labels: See General Comment 1.

Response: revised. **New text line 618-627.**